# Winformer: Transcending Pairwise Similarity for Time-series Generation

**Haoyi Zhou**[1]  **Xin Xue**[1]  **Tianyu Chen**[1]  **Lanhao Li**[1]  **Lijun Sun**[1]  **Jianxin Li**[1]

## Abstract

The periodicity misalignment remains a challenge problem in generating time-series data across multiple domains. The fundamental processing unit of attention in time-series modeling has long been restricted to either individual points or fragmented segments, limiting their ability to capture and adapt to complex periodic patterns inherent in diverse domains. To address this, we introduce Winformer, first to extend this processing unit from individual points to sliding windows, establishing a unified window-wise attention paradigm. Leveraging the adaptive window-alignment kernels derived from the frequency decomposition, Winformer brings semantically richer window representations, and effectively captures and transfers complex periodic patterns across domains. Extensive experiments on 12 real-world datasets demonstrate Winformer's effectiveness, achieving an average performance gain of 10.67% over SOTA baselines.

## 1. Introduction

When generating data across multiple time-series domains, one of the major issue is the *domain confusion* (Liu et al., 2024) that different domains exhibits diverse temporal distribution. The normalization techniques (Kim et al., 2022; Ulyanov et al., 2016; Ogasawara et al., 2010) could alleviate the magnitude misalignment, and we focus on the phase misalignment (Zhang et al., 2025; Yuan & Qiao, 2024), i.e. periodicity, which may arise from sampling rates, system delays, manual operations, or inconsistencies environment.

Existing approaches models the periodicity in point-wise or patch-wise, both of which fall short in addressing domain confusion. Following the vanilla Transformer design (Vaswani et al., 2017; Zhou et al., 2021), the previous methods (Yuan & Qiao, 2024; Piao et al., 2024; Zhou et al.,

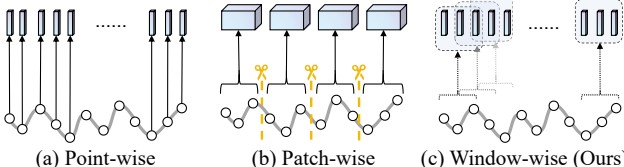

*Figure 1.* **The perspective of time-series modeling.** (a) Point-to-point modeling on exact interactions (**Slicing** window, window size=1). (b) Patch-to-patch modeling on middle-level interactions (**Slicing** window, window size=3). (c) We propose window-to-window modeling on complex interactions (**Sliding** window, window size=3).

2022) model the exact interactions between individual time-series through a point-wise perspective, as illustrated in Fig. 1(a). With limited representation ability, these methods require additional frequency components to capture long-range and high-order interactions (Zhou et al., 2022) and stacking of attention layers (Wu et al., 2021). Recently, the time-series patching (Nie et al., 2023; Peebles & Xie, 2023) is introduced to obtain middle-level interactions. However, the equidistant patching directly breaks evolving trends and periodic patterns, as in Fig. 1(b). Thus, how to adaptively capture the periodicity across diverse domains remains a challenge in cross-domain time-series generation.

To tackle this critical limitation, we are the first to extend the fundamental principle of attention mechanism, shifting from pairwise point similarity to continuous window comparison. To address this issue, we extend the fundamental principle of attention mechanism, shifting the pairwise point similarity, to windows comparison. As shown in Fig. 1, in point-wise modeling, the slicing window is set as 1 as the attention calculates on pairwise points, while in patch-wise modeling, the slicing window is set as 3 and the attention is forced on pairwise lattices. Unlike the fixed slicing windows used in traditional data processing, our approach provides a new way by introducing the sliding window comparison during attention calculation shown in Fig. 1(c), enabling adaptive periodicity alignment of across domains. Specifically, we propose the Ample attention, which adaptively captures the periodicity with window-wise aligning kernel based on frequency decomposition, enabling the window-wise similarity comparison. With the periodicity aligning across domains, our method effectively enhances domain adaptability thereby improves the generation quality.

[1]Beihang University, Beijing, China. Correspondence to: Jianxin Li <lijx@buaa.edu.cn>.

*Proceedings of the 43rd International Conference on Machine Learning*, Seoul, South Korea. PMLR 306, 2026. Copyright 2026 by the author(s).

To summarize, Our contributions are:

- We are the first to rethink the fundamental processing unit of attention for time series and extend it from pairwise point similarity to sliding window comparison. We establish a unified window-wise attention paradigm, providing a principled foundation for periodicity-aware cross-domain generation.

- We reveal a fundamental equivalence between window-wise similarity in the spectral domain and learnable convolutions on the original attention score map, bridging frequency decomposition with convolutional attention. This theoretical result leads to the Ample attention, an efficient mechanism that captures periodic alignment without explicit Fourier transforms.

- Extensive experiments on 12 real-world datasets across four domains demonstrate the effectiveness of the proposed paradigm. Winformer achieves state-of-the-art performance with an average improvement of 10.67% in MMD, and visualization analysis confirms that the window-wise mechanism recovers finer periodic structures during the denoising process.

## 2. Related work

Existing time-series generation models mainly consist of GAN-based models, VAE-based models and diffusion-based models. GAN-based models (Yoon et al., 2019; Jeon et al., 2022) possess advantages of not requiring distribution assumptions, but show poor training stability. VAE-based models (Desai et al., 2021; Lee et al., 2023) can achieve training stability with clear optimization target, but show limited generation quality and diversity. Diffusion models (Huang et al., 2025; Ge et al., 2025), which are popular paradigms, have advantages in both training stability and generation quality.

Transformer-based diffusion models show a more powerful effect in various time-series tasks, including forecasting (Feng et al., 2024), anomaly detection (Wang & Li, 2025) imputation (Alcaraz & Strodthoff, 2023; Tashiro et al., 2021), and generation (Peebles & Xie, 2023; Yuan & Qiao, 2024; Cao et al., 2025). In addition to domain-specific modeling, there are also transformer-based diffusion models for cross-domain generation (Huang et al., 2025; Ge et al., 2025; Ma et al., 2025). However, the these methods lack of an alignment mechanism for domain-wise correlations, which require further exploration.

## 3. Preliminary

With the rolling forecasting setting with a fixed horizon, we define the $t$-th sequence inputs of domain $j$ as $\mathcal{X}^{(j,t)} =$

$\{\mathbf{x}_1^{(j,t)}, \ldots, \mathbf{x}_{L_x}^{(j,t)} \mid \mathbf{x}_i^{(j,t)} \in \mathbb{R}^D\}$, where $L_x$ stands for the horizon length. We mix the $M$ datasets from different domains together to get the unified dataset $\mathcal{X} = \cup_{i=1}^M \mathcal{X}_i$ considering the cross-domain time-series generation settings, and the overall target is to learn a parameterized model $\theta$ with $p_\theta(\mathbf{x}_1, \mathbf{x}_2, ..., \mathbf{x}_T | i)$ for the $i$-th dataset.

### 3.1. Vanilla Self-attention

The self-attention mechanism (Vaswani et al., 2017) yields successful pairwise alignment ability in sequence modeling. It is calculated on three transformed inputs from $\mathbf{X} \in \mathbb{R}^{L_x \times D}$, i.e, query, key and value, which is defined as the scaled dot-product as:

$$\mathcal{A}(\mathbf{Q}, \mathbf{K}, \mathbf{V}) = \text{Softmax}(\frac{\mathbf{Q}\mathbf{K}^\top}{\sqrt{d}})\mathbf{V} \qquad , \qquad (1)$$

where we have $\mathbf{Q} = \mathbf{X}\mathbf{W}_Q^\top$, $\mathbf{K} = \mathbf{X}\mathbf{W}_K^\top$, $\mathbf{V} = \mathbf{X}\mathbf{W}_V^\top$ with trivial projections $\mathbf{Q} \in \mathbb{R}^{L_Q \times d}$, $\mathbf{K} \in \mathbb{R}^{L_K \times d}$, $\mathbf{V} \in \mathbb{R}^{L_V \times d}$ and the efficient scaled norm of the dimension $d$.

### 3.2. Denoising Diffusion Probabilistic Models

The diffusion probabilistic generation methods are based on the assumption of Markov chain. Specifically, let the input matrix $\mathbf{X}_{(0)} \in \mathbb{R}^{L_x \times D} \sim q(\mathbf{X})$ be the real data. At the $n$ step of the diffusion process, we obtain $\mathbf{X}_{(n)}$ from $\mathbf{X}_{(n-1)}$ by adding Gaussian noise with the transition kernel defined as $q(\mathbf{X}_{(n)}|\mathbf{X}_{(n-1)}) = \mathcal{N}(\mathbf{X}_{(n)}; \sqrt{1-\beta_n}\mathbf{X}_{(n)}, \beta_n\mathbf{I})$, where $\beta_N \in (0,1)$. By recurring the $n$ steps of the Markov chain, we can derive:

$$q(\mathbf{X}_{(n)}|\mathbf{X}_{(0)}) = \mathcal{N}\left(\mathbf{X}_{(n)}; \sqrt{\bar{\alpha}_n}\mathbf{X}_{(n)}, (1-\bar{\alpha}_n)\mathbf{I}\right), \quad (2)$$

where $\alpha_n = 1 - \beta_n$ and $\bar{\alpha}_n = \prod_{i=1}^n \alpha_i$. In reversing, a learning-based model reconstructs as

$$p_\theta(\mathbf{X}_{(n-1)}|\mathbf{X}_{(n)}) = \mathcal{N}\left(\mathbf{X}_{(n-1)}; \mu_\theta(\mathbf{X}_{(n)}, n), \Sigma_\theta\right), \quad (3)$$

where $\mu_\theta(\mathbf{X}_{(n)}, n)$ is estimated as

$$\mu_\theta(\mathbf{X}_{(n)}, n) = \frac{1}{\sqrt{\alpha_n}}\left(\mathbf{X}_{(n)} - \frac{\beta_n}{\sqrt{1-\bar{\alpha}_n}}\epsilon_\theta(\mathbf{X}_{(n)}, n)\right). \quad (4)$$

## 4. Methods

In this section, we introduce the Winformer, a diffusion model with window-wise alignment as illustrated in Fig. 2.

### 4.1. Theoretical Analysis for Window-wise Alignment

The vanilla self-attention mechanism is built upon the point-to-point dot-product similarity. It achieves significant alignment ability over the tokenized inputs of language (Devlin et al., 2019), where the text embedding Word2Vec (Mikolov

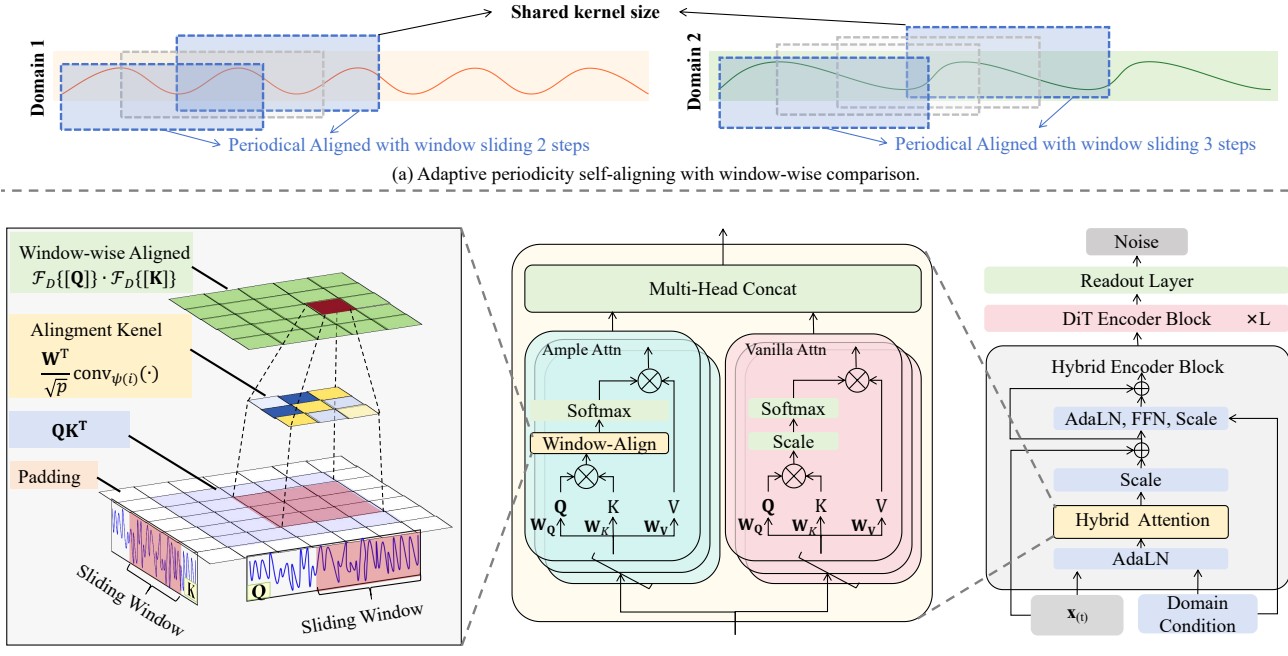

(a) Adaptive periodicity self-aligning with window-wise comparison.

(b) Window-wise Alignment.  (c) Ample Attention for Hybrid Encoding.  (c) Denoising with Domain Conditon.

*Figure 2.* **The overall architecture. (a) Periodicity Aligning with sliding windows:** Adaptive window-wise alignment allows periodicity self-aligning across diverse domains by sliding kernel with shared kernel size. **(b) Window-wise Alignment:** Based on the frequency decomposition, the window-wise alignment is realized by a convolutional kernel sliding on similarity maps. **(c) Ample Attention for Hybrid Encoding.:** The hybrid attention concatenate the heads of Ample attention and vanilla attention for feature fusing. **(d) Denoising with Domain Condition:** With domain conditioning, the denoising network predicts noise for cross-domain time-series generation.

et al., 2013) and Jina (Günther et al., 2023) ensure the unified linguistic space. For time-series generation, the Fourier transform decomposes the sequences into constituent frequencies and defines similarity between the frequency coefficients to make it possible to align the series. Building on this, we propose transforming time-series data into the Fourier basis and calculating similarity over complex planes.

### 4.1.1. FREQUENCY DECOMPOSITION OVER SIMILARITY

Recalling that we have an input matrix $\mathbf{X}$, the $i$-th time step is represented by a vector $\mathbf{x}_i \in \mathbb{R}^{1 \times D}$. Assume that we perform a window-wise Fourier transformation, over these finite-length signals, and the window size is set as $p$. Since the time-series become discrete-time inputs, we use the discrete Fourier transform (DFT) $\mathcal{F}_D$ instead. We apply the DFT operator to each feature dimension independently, which follows the same setting (Alaa et al., 2021). We select the $q$-th dimension of projected inputs $\mathbf{Q}$ as $\mathbf{Q}^q = \{\mathbf{x}_1^q, \ldots, \mathbf{x}_{L_x}^q\}$. Through the temporal zero padding on the beginning and ending, we have the window-wise attention inputs $\mathbf{Q}^q$ and $\mathbf{K}^q$ respectively:

$$[\mathbf{Q}]_i^{(q)} = \{\mathbf{Q}_j^q | i \times s \le j < (i+1) \times s\} \quad , \quad (5)$$

and

$$[\mathbf{K}]_i^{(q)} = \{\mathbf{K}_j^q | i \times s \le j < (i+1) \times s\} \quad , \quad (6)$$

where the $[\mathbf{Q}]_i^{(q)} \in \mathbb{R}^{s \times 1}$, $[\mathbf{K}]_i^{(q)} \in \mathbb{R}^{s \times 1}$ and $s$ stands for the stride. For the $t$-th time step, the DFT transforms the real-valued inputs into the complex-valued ones as $\mathcal{F}_D\{[\mathbf{Q}]_t^{(q)}\}$ and $\mathcal{F}_D\{[\mathbf{K}]_t^{(q)}\}$. Likewise the self-attention mechanism, we can perform the dot-product between the selected time step $t_1$ and $t_2$ in the frequency domain, whose scores calculate the similarity between different spectral components (Zhou et al., 2022; Piao et al., 2024; Kong et al., 2023) as:

$$\widetilde{\mathbf{F}}_{(t_1,t_2)}^{(p,q)} = \mathcal{F}_D\{[\mathbf{Q}]_{t_1}^{(q)}\} \cdot \mathcal{F}_D\{[\mathbf{K}]_{t_2}^{(q)}\} \quad . \quad (7)$$

This score is a complex matrix containing the real component $\mathbf{Re}(\widetilde{\mathbf{F}})$ and imaginary component $\mathbf{Im}(\widetilde{\mathbf{F}})$, a two-channel matrix with shape $2 \times s \times 1$. The former component measures the magnitude difference between the spectral coefficients, while the latter one is about the phase difference.

Then, we can concatenate all the inputs' DFT result as $\widetilde{\mathbf{F}}_{(t_1,t_2)}^{(p)} = [\widetilde{\mathbf{F}}_{(t_1,t_2)}^{(p,1)}, \ldots, \widetilde{\mathbf{F}}_{(t_1,t_2)}^{(p,d)}]$ with a shape of $2 \times s \times d$. Likewise the self-attention mechanism, we take the magnitude difference of the spectral coefficients at $t_1$ and $t_2$ as an example, which could be normalized by Softmax($\cdot$) to reorganize $\mathbf{V}$. Then we can define the real-valued similarity within the window-to-window comparison as:

$$\mathbf{S}_{(t_1,t_2)}^{\mathbf{Re}} = \mathbf{Re}(\widetilde{\mathbf{F}}_{(t_1,t_2)}^{(p)})\mathbf{1} \quad , \quad (8)$$

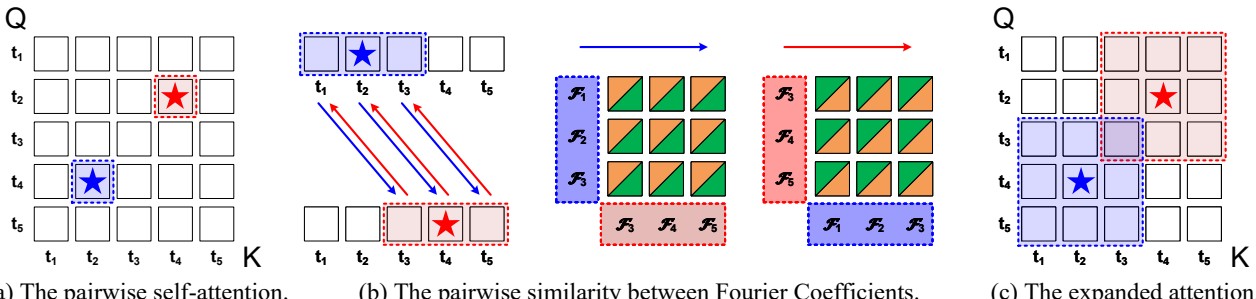

(a) The pairwise self-attention.     (b) The pairwise similarity between Fourier Coefficients.     (c) The expanded attention.

*Figure 3.* The process of attention expansion. (a) The visualization of pairwise feature map (dot-product) of vanilla self-attention. There are five tokens $(t_1, t_2, t_3, t_4, t_5)$ in both query and keys. And the blue star $(Q_{t_4}, K_{t_2})$ and red star $(Q_{t_2}, K_{t_4})$ form a duality. (b) The new similarity is defined between the windowed Fourier coefficients. We set the window size as 3, and the matrix is the dot-product of corresponding coefficients, where the orange triangle stands for real parts and the green triangle for imaginary parts. The blue arrow denotes the grouped similarity from $(t_1, t_2, t_3)$ to $(t_3, t_4, t_5)$ like the blue star in figure a. (c) We design the expanded attention to crop the scores from original feature map.

where $\mathbf{1} \in \mathbb{R}^{d \times 1}$ denotes the all-one vector and it sums the real coefficient of all channels (also applies to imaginary).

As illustrated in Fig. 3, the original self-attention leverages the pairwise similarity to measure the relationship between different inputs, e.g., the blue star represents the attention feature map for $Q_{t_4}$ and $K_{t_2}$ pairs and red star is the dual score. If we perform the windowed DFT and measure the "group" similarity between $(t_1, t_2, t_3)$ and $(t_3, t_4, t_5)$ through the real-valued similarity defined in Eq.(8) and its imaginary ones, we can acquire the blue arrows composed with real parts (the orange triangle) and imaginary parts (the green triangle). Based on the following derivation, we will demonstrate that the pairwise attention could be expanded to crop scores, where the score (blue star) stands for window-to-window comparison.

### 4.1.2. THE CONVOLUTIONAL CALCULATION

The Fourier Transform, a mathematical operator (Duhamel & Vetterli, 1990), can be calculated in Eq.(7), but it requires two transformations, namely product in frequency domain and composition of real part and imaginary part. Actually, our ultimate goal is to calculate a new similarity score in window-wise perspective that is numerically compatible with other network components. Inspired by the convolution theorem (McGillem & Cooper, 1991), which bridges the connection between Fourier operator and convolution operator, we decompose the calculation of the DFT operator as a linear transformation:

$$\mathcal{F}_D(\mathbf{x}) = \mathbf{M}\mathbf{x} \quad , \text{ where}$$

$$\mathbf{M} = \frac{1}{\sqrt{p}} \begin{bmatrix} 1 & 1 & 1 & \cdots & 1 \\ 1 & \omega & \omega^2 & \cdots & \omega^{p-1} \\ 1 & \omega^2 & \omega^4 & \cdots & \omega^{2(p-1)} \\ \vdots & \vdots & \vdots & \ddots & \vdots \\ 1 & \omega^{p-1} & \omega^{2(p-1)} & \cdots & \omega^{(p-1)^2} \end{bmatrix} . \quad (9)$$

The coefficient $\omega$ is $e^{-2\pi j/p}$. Using the Euler's rule and the $i$-th row of transformation $\mathbf{M}$ be formulated as $\psi(i) = [1, \cos(2\pi i/p), \dots, \cos(2\pi i(p-1)/p)]$, we can rewrite $\mathbf{M}$ as $[\psi(0), \dots, \psi(p-1)]^\top/\sqrt{p}$, and the window-wise similarity score is the sum of different groups:

$$\mathbf{S}_{(t_1, t_2)} = \frac{\mathbf{W}^\top}{\sqrt{p}} \left[ \begin{pmatrix} \psi(0) \\ \vdots \\ \mathbf{0} \end{pmatrix} + \cdots + \begin{pmatrix} \mathbf{0} \\ \vdots \\ \psi(p-1) \end{pmatrix} \right] \begin{pmatrix} \varphi_0 \\ \vdots \\ \varphi_{p-1} \end{pmatrix}, \quad (10)$$

where the $\mathbf{W}$ is the coefficient for decomposed and reorganized $p$ sub-matrix and the $\mathbf{0}$ is the all zero matrix.

We noticed that the pairwise attention scores $\mathbf{S}'_{(t_1, t_2)}$ has been already calculated in Eq.(1). The target window-wise score $\mathbf{S}_{(t_1, t_2)}$ can be acquired by performing convolution whose shape is larger than or equal to the periodic one in Eq.(10). Let $\varphi = [\varphi_0, \varphi_1, \dots, \varphi_{p-1}]$, we have:

$$\begin{aligned} \mathbf{S}_{(t_1, t_2)} &= \frac{\mathbf{W}^\top}{\sqrt{p}} \left[ \psi(0)\varphi + \cdots + \psi(p-1)\varphi \right] \\ &= \frac{\mathbf{W}^\top}{\sqrt{p}} [\text{avg}(\mathbf{S}') + \text{conv}_{\psi(1)}(\mathbf{S}') + \cdots + \text{conv}_{\psi(p-1)}(\mathbf{S}')] \end{aligned}, \quad (11)$$

where $\text{conv}_{\psi(i)}(\cdot)$ represents the convolution operator with the kernel $\psi(i)$, with the same basis of DFT in Eq.(9). Thus, we can acquire the window-wise score by performing the convolution operator on the original attention score. Specially, for $\psi(0)$, we can use an all-one matrix as the kernel, which equals that we only select the $\text{avg}(\cdot)$ operator.

### 4.2. Establishing Ample Attention for Hybrid Encoding

Given the window-wise alignment defined on the similarity as Eq.(11), we introduce the Ample attention with alignment kernel and incorporate it with the vanilla attention to blend the window-wise and point-wise features. Specially, for the hybrid encoder block, we divide the attention heads

into two groups, including the vanilla attention heads and the Ample attention heads, and concatenate the results from these heads. The vanilla attention heads preserve point-wise similarity in Eq.(1), and the Ample attention is calculated with the alignment kernel formally defined as following.

### 4.2.1. THE AMPLE ATTENTION

Recalling the vanilla Transformer's attention mechanism, we still follows the Softmax($\cdot$) design while replacing the pairwise similarity with the window-wise comparison. We applied generalized Parseval's theorem (Hardy & Titchmarsh, 1931) between the pairwise $\mathbf{S}'_{(t_1, t_2)}$ and window-wise $\mathbf{S}_{(t_1, t_2)}$, it reveals a linear connection on the sum of scores. It motivates us to leverage the Conv2d($\cdot$) layer with learnable kernel, we initialized the kernel with a decomposing basis for fast convergence, then the distribution of linear coefficients $\mathbf{W}$ is learned through convolution kernel. In this way, we can define the Ample Attention as:

$$\text{Attn} = \text{Softmax}\left(\text{Conv2d}_\psi(\mathbf{Q}\mathbf{K}^{\mathrm{T}})\mathbf{V}\right) \qquad , \qquad (12)$$

where the kernel $\psi$ could be initialized by an exact type of transformation. If we only consider the term of $\psi(0)$ in Eq.( 11), we can perform Avgpool2d($\cdot$) to replace the Conv2d$_\psi(\cdot)$.

### 4.2.2. THE REDUCED KERNEL

In practice, as the Fourier basis involves imaginary calculation, bring difficulties for calculation, we simplify the window-wise alignment kernel with a Discrete Cosine Transform (DCT), which is a reduced form of DFT, containing real parts only. The discussion of the relationship and theoretical proofs are provided in Appendix I.5. Specifically, as As the previous derivation along with Eq.(11), we can initialize the kernel of Conv2d$_\psi(\cdot)$ with the DCT basis, which is depicted as

$$\psi(i,j) = \begin{cases} \sqrt{\dfrac{1}{p}} \cos[\dfrac{(j + 0.5\pi)}{p}i] & i = 0 \\ \sqrt{\dfrac{2}{p}} \cos[\dfrac{(j + 0.5\pi)}{p}i] & i \neq 0 \end{cases} . \quad (13)$$

Because of the learnable ability of the convolutional operators, the kernel will further adapt window alignment. With the reduced convolutional kernel, the ample attention calculates in a efficient way, and further computation costs are shown in Appendix C.

### 4.3. Denoising and Domain Conditioning

Integrating with the hybrid encoder, we establish the denoising network for diffusion process. As Eq.( 3), for the $n$-th diffusion step, the denoising network iteratively predict the noise $\epsilon_\theta$ from the noisy data $\mathbf{X}_{(n)}$.

To facilitate cross-domain generation, we construct the domain prompts inspired by TimeDP (Huang et al., 2025). Specifically, we construct the condition $c$ with domain prompt $c_{\mathrm{d}}$, learned prototypes $c_{\mathrm{p}}$ and step information $c_{\mathrm{t}}$. And it is defined as

$$c = \text{Emb}_{\mathrm{t}}(c_{\mathrm{t}}) + \text{Emb}_{c_{\mathrm{p}}}(c_{\mathrm{p}}) + \text{Emb}_{c_{\mathrm{d}}}(c_{\mathrm{d}}) \qquad , \qquad (14)$$

where $\text{Emb}_{\mathrm{t}}(\cdot)$, $\text{Emb}_{c_{\mathrm{p}}}(\cdot)$ and $\text{Emb}_{c_{\mathrm{d}}}(\cdot)$ represent the embedding methods for these information. The condition $c$ is injected by an adaptive Layer Norm Function as introduced by DiT (Peebles & Xie, 2023).

Injected with the condition $c$, the $\mathbf{X}_{(n)}$ is processed by a stack of encoders and readout layer, which help to reconstruct $\mathbf{X}_{(n-1)}$ from $\mathbf{X}_{(n)}$.

## 5. Experiments

In this section, we empirically demonstrate the effectiveness of Winformer on the 12 real-world datasets and further discuss its generating process through visualization. We also provide more experimental details and additional results in Appendix A to Appendix E, including hyper-parameter tests, computational costs, visualization figures and domain discrepancy analysis.

### 5.1. Setup: Cross-domain Time-series Generation

**(a) Datasets.** We conduct experiments on 12 real-world time-series datasets across four domains following the setting of TimeDP (Huang et al., 2025), including the time-series data of traffic flows, weather phenomena, industrial logs and financial records. All these datasets are reformatted into non-overlapping univariate sequence slices with the fixed length of 168. For cross-domain generation, time-series from diverse domains are mixed during training.

**(b) Baselines.** We compare our model with 6 representative SOTA methods for cross-domain time-series generation. These methods include GAN-base methods, such as TimeGAN (Yoon et al., 2019) and GT-GAN (Jeon et al., 2022), VAE-based methods, such as TimeVAE (Desai et al., 2021) and TimeVQVAE (Lee et al., 2023), and diffusion-based method, such as the newly released TimeDP (Huang et al., 2025) and Diffussion-TS (Yuan & Qiao, 2024). To ensure a fair comparison, we adopt partial related results reported by TimeDP (Huang et al., 2025).

**(c) Metrics.** We select two main metrics to evaluate the performance of generation by measuring the similarity between the distributions of the real and generated time-series. **Maximum Mean Discrepancy (MMD)** compares the discrepancy between two series after mapping into a high-dimensional feature space with a kernel function. **Kullback-Leibler Divergence (K-L)** measures the divergence between two probability distributions. The metric computation

*Table 1.* Results of generation results for sequence length 168. Best results are **bold** and second best results are underlined. Our method outperforms SOTA baselines in most of the datasets and achieve an increase of averagely 10.67% in MMD.

| | | Winformer | TimeDP | Diffusion-TS | TimeGAN | GT-GAN | TimeVAE | TimeVQVAE |
|---|---|---|---|---|---|---|---|---|
| Maximum Mean Discrepancy | Electricity | $\mathbf{0.001}_{\pm 0.002}$ | $\mathbf{0.001}_{\pm 0.001}$ | $0.003_{\pm 0.002}$ | $0.367_{\pm 0.255}$ | $0.254_{\pm 0.166}$ | $0.577_{\pm 0.006}$ | $0.152_{\pm 0.024}$ |
| | Solar | $\mathbf{0.035}_{\pm 0.004}$ | $\underline{0.041}_{\pm 0.011}$ | $0.050_{\pm 0.012}$ | $0.628_{\pm 0.053}$ | $0.578_{\pm 0.039}$ | $0.353_{\pm 0.014}$ | $0.437_{\pm 0.020}$ |
| | Wind | $\underline{0.034}_{\pm 0.014}$ | $\mathbf{0.025}_{\pm 0.017}$ | $0.035_{\pm 0.006}$ | $0.213_{\pm 0.017}$ | $0.170_{\pm 0.040}$ | $0.170_{\pm 0.004}$ | $0.131_{\pm 0.014}$ |
| | Traffic | $\mathbf{0.071}_{\pm 0.005}$ | $\underline{0.083}_{\pm 0.034}$ | $0.111_{\pm 0.031}$ | $0.567_{\pm 0.057}$ | $0.538_{\pm 0.078}$ | $0.218_{\pm 0.007}$ | $0.213_{\pm 0.016}$ |
| | Taxi | $\mathbf{0.085}_{\pm 0.010}$ | $\underline{0.095}_{\pm 0.023}$ | $0.131_{\pm 0.014}$ | $0.275_{\pm 0.054}$ | $0.319_{\pm 0.032}$ | $0.139_{\pm 0.007}$ | $0.128_{\pm 0.004}$ |
| | Pedestrian | $\mathbf{0.040}_{\pm 0.008}$ | $\underline{0.044}_{\pm 0.020}$ | $0.071_{\pm 0.019}$ | $0.090_{\pm 0.030}$ | $0.112_{\pm 0.019}$ | $0.065_{\pm 0.002}$ | $0.067_{\pm 0.007}$ |
| | Air | $\mathbf{0.011}_{\pm 0.002}$ | $\mathbf{0.011}_{\pm 0.003}$ | $\underline{0.022}_{\pm 0.011}$ | $0.120_{\pm 0.045}$ | $0.211_{\pm 0.041}$ | $0.089_{\pm 0.016}$ | $0.028_{\pm 0.002}$ |
| | Temperature | $\underline{0.230}_{\pm 0.021}$ | $\mathbf{0.219}_{\pm 0.022}$ | $0.241_{\pm 0.049}$ | $0.926_{\pm 0.042}$ | $0.809_{\pm 0.081}$ | $1.002_{\pm 0.014}$ | $0.323_{\pm 0.008}$ |
| | Rain | $\mathbf{0.036}_{\pm 0.016}$ | $0.057_{\pm 0.039}$ | $0.079_{\pm 0.058}$ | $0.329_{\pm 0.285}$ | $0.111_{\pm 0.109}$ | $0.292_{\pm 0.019}$ | $\underline{0.074}_{\pm 0.007}$ |
| | NN5 | $\mathbf{0.147}_{\pm 0.008}$ | $\underline{0.164}_{\pm 0.010}$ | $0.186_{\pm 0.043}$ | $0.874_{\pm 0.088}$ | $0.632_{\pm 0.074}$ | $0.821_{\pm 0.061}$ | $0.327_{\pm 0.012}$ |
| | Fred-MD | $\mathbf{0.002}_{\pm 0.001}$ | $\mathbf{0.002}_{\pm 0.001}$ | $0.006_{\pm 0.002}$ | $0.043_{\pm 0.021}$ | $0.133_{\pm 0.102}$ | $0.059_{\pm 0.008}$ | $\underline{0.008}_{\pm 0.002}$ |
| | Exchange | $\mathbf{0.137}_{\pm 0.012}$ | $\underline{0.151}_{\pm 0.024}$ | $0.206_{\pm 0.035}$ | $0.530_{\pm 0.154}$ | $0.475_{\pm 0.116}$ | $0.543_{\pm 0.149}$ | $0.342_{\pm 0.050}$ |
| K-L Divergence | Electricity | $\mathbf{0.008}_{\pm 0.010}$ | $\underline{0.012}_{\pm 0.016}$ | $0.315_{\pm 0.247}$ | $0.488_{\pm 0.175}$ | $0.407_{\pm 0.079}$ | $0.734_{\pm 0.023}$ | $0.280_{\pm 0.051}$ |
| | Solar | $\mathbf{0.013}_{\pm 0.005}$ | $\underline{0.016}_{\pm 0.005}$ | $0.066_{\pm 0.055}$ | $0.612_{\pm 0.447}$ | $0.120_{\pm 0.041}$ | $0.260_{\pm 0.016}$ | $0.865_{\pm 0.108}$ |
| | Wind | $0.202_{\pm 0.044}$ | $\underline{0.152}_{\pm 0.034}$ | $0.548_{\pm 0.372}$ | $1.924_{\pm 1.233}$ | $\mathbf{0.107}_{\pm 0.016}$ | $0.484_{\pm 0.015}$ | $0.483_{\pm 0.066}$ |
| | Traffic | $\underline{0.011}_{\pm 0.002}$ | $\mathbf{0.009}_{\pm 0.003}$ | $0.120_{\pm 0.074}$ | $1.305_{\pm 0.320}$ | $1.409_{\pm 0.251}$ | $0.211_{\pm 0.014}$ | $0.178_{\pm 0.026}$ |
| | Taxi | $\mathbf{0.005}_{\pm 0.003}$ | $0.011_{\pm 0.004}$ | $0.075_{\pm 0.034}$ | $0.650_{\pm 0.180}$ | $0.950_{\pm 0.197}$ | $\underline{0.110}_{\pm 0.020}$ | $0.110_{\pm 0.026}$ |
| | Pedestrian | $\mathbf{0.009}_{\pm 0.004}$ | $\underline{0.014}_{\pm 0.010}$ | $0.133_{\pm 0.069}$ | $0.417_{\pm 0.181}$ | $0.411_{\pm 0.096}$ | $0.065_{\pm 0.005}$ | $0.405_{\pm 0.051}$ |
| | Air | $\mathbf{0.026}_{\pm 0.011}$ | $\underline{0.027}_{\pm 0.016}$ | $0.106_{\pm 0.079}$ | $0.348_{\pm 0.093}$ | $0.578_{\pm 0.049}$ | $0.164_{\pm 0.012}$ | $0.054_{\pm 0.012}$ |
| | Temperature | $\underline{0.176}_{\pm 0.027}$ | $\mathbf{0.171}_{\pm 0.073}$ | $0.342_{\pm 0.131}$ | $8.892_{\pm 2.681}$ | $3.174_{\pm 2.685}$ | $2.183_{\pm 0.110}$ | $0.735_{\pm 0.066}$ |
| | Rain | $\mathbf{0.011}_{\pm 0.003}$ | $\underline{0.013}_{\pm 0.012}$ | $0.061_{\pm 0.072}$ | $0.506_{\pm 0.174}$ | $0.432_{\pm 0.099}$ | $0.160_{\pm 0.022}$ | $0.047_{\pm 0.018}$ |
| | NN5 | $\mathbf{0.045}_{\pm 0.007}$ | $\underline{0.054}_{\pm 0.014}$ | $0.165_{\pm 0.076}$ | $4.928_{\pm 4.112}$ | $1.386_{\pm 0.520}$ | $1.337_{\pm 0.220}$ | $1.063_{\pm 0.274}$ |
| | Fred-MD | $\mathbf{0.201}_{\pm 0.014}$ | $\underline{0.203}_{\pm 0.035}$ | $0.835_{\pm 0.554}$ | $0.512_{\pm 0.290}$ | $0.380_{\pm 0.070}$ | $0.346_{\pm 0.041}$ | $0.831_{\pm 0.077}$ |
| | Exchange | $\mathbf{1.621}_{\pm 0.126}$ | $\underline{1.866}_{\pm 0.132}$ | $2.337_{\pm 0.714}$ | $8.861_{\pm 3.397}$ | $7.201_{\pm 4.380}$ | $10.404_{\pm 1.434}$ | $5.052_{\pm 1.385}$ |
| Count | | 19 | 7 | 0 | 0 | 1 | 0 | 0 |

is stated in Appendix G.1. MMD measures global distribution, while KL is sensitive to local point-wise noise. These two metrics measure the results in different aspect. In addition to intra-domain comparison, we also evaluated the domain discrepancy for inter-domain comparison, with full results shown in Appendix E.

**(d) Implementation.** As analyzed in Section 4, the stride length of window-alignment is a hyper-parameter, and we choose 25 in our main experiments, which is suitable due to it is larger than common periodic cycles in real-world datasets, such as 4, 6, 12 and 24. We further explored the effectiveness with different stride length and kernel type of window-alignment in Appendix A. The number of DiT encoder layers is set to 6, and the hidden size is 512. The learning rate is set to $5 \times 10^{-5}$ with $1,000$ warm-up steps. For the diffusion process, we use 200 steps adding noise to the series or reconstructing them. The experiments are repeated 5 times with seeds spanning from 2021 to 2025. More details about implementation can be found in the Appendix G.

### 5.2. Main Results

In this experiment, we evaluate our methods with 6 representative baselines in Table 1. Our method achieves the best performance in 10 out of the 12 datasets measuring with maximum mean discrepancy, which indicates that the time-series generated by our model conforms to the original sequence better than other competitive methods. Considering all of the 12 datasets, our method averagely decreases the MMD with 10.67% compared to TimeDP, which is the SOTA method of cross-domain generation.

From the detailed results on each domain, we can find that the Winformer demonstrates especially stronger ability on datasets with strong periodicity, such as solar and traffic. On the contrary, the ability of Winformer lags slightly behind on datasets exhibiting a stronger tendency toward trendiness rather than periodicity, such as wind and temperature. For further analysis on periodicity capturing, we visualized an example of the generated series with both the original signal and the frequency spectrum as shown in Fig. 4. It's obvious that time-series signals generated by our method are more

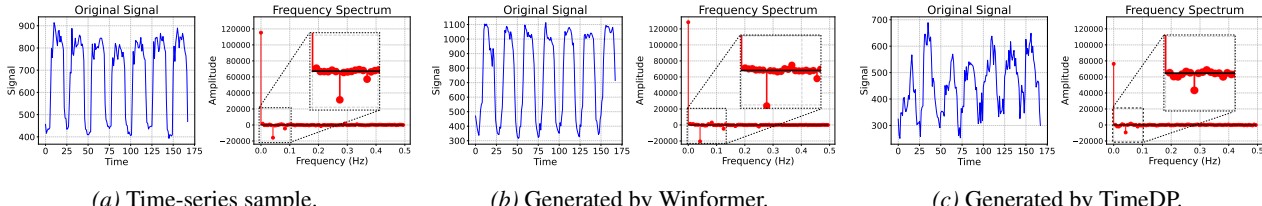

*(a)* Time-series sample.    *(b)* Generated by Winformer.    *(c)* Generated by TimeDP.

*Figure 4.* Visualization of generated series, comparing on original signal and frequency spectrum. Through comparison, we found that the samples generated by our model exhibit better feature learning in the frequency domain.

*Table 2.* Results for ablation study on point-wise, patch-wise and our window-wise alignment. The best results of each line are **bold**, and our window-wise alignment outperforms other methods with 15 best scores.

| Alignment Type | Window-wise Alignment | | Patch-wise Alignment | | Point-wise Alignment | |
|---|---|---|---|---|---|---|
| Metric | MMD | K-L | MMD | K-L | MMD | K-L |
| Electricity | $\mathbf{0.001}_{\pm 0.002}$ | $\mathbf{0.008}_{\pm 0.010}$ | $0.002_{\pm 0.005}$ | $0.011_{\pm 0.005}$ | $0.002_{\pm 0.004}$ | $0.030_{\pm 0.028}$ |
| Solar | $\mathbf{0.035}_{\pm 0.004}$ | $\mathbf{0.013}_{\pm 0.005}$ | $\mathbf{0.035}_{\pm 0.003}$ | $0.016_{\pm 0.007}$ | $\mathbf{0.035}_{\pm 0.003}$ | $0.017_{\pm 0.011}$ |
| Wind | $0.034_{\pm 0.014}$ | $0.202_{\pm 0.044}$ | $\mathbf{0.033}_{\pm 0.010}$ | $\mathbf{0.176}_{\pm 0.005}$ | $0.034_{\pm 0.010}$ | $0.186_{\pm 0.026}$ |
| Traffic | $\mathbf{0.071}_{\pm 0.005}$ | $0.011_{\pm 0.002}$ | $0.109_{\pm 0.012}$ | $0.012_{\pm 0.010}$ | $0.073_{\pm 0.008}$ | $\mathbf{0.010}_{\pm 0.003}$ |
| Taxi | $\mathbf{0.085}_{\pm 0.010}$ | $0.005_{\pm 0.003}$ | $0.195_{\pm 0.000}$ | $0.014_{\pm 0.000}$ | $0.088_{\pm 0.011}$ | $\mathbf{0.004}_{\pm 0.002}$ |
| Pedestrain | $0.040_{\pm 0.008}$ | $\mathbf{0.009}_{\pm 0.004}$ | $\mathbf{0.034}_{\pm 0.014}$ | $0.011_{\pm 0.135}$ | $0.041_{\pm 0.010}$ | $0.012_{\pm 0.009}$ |
| Air | $\mathbf{0.011}_{\pm 0.002}$ | $\mathbf{0.026}_{\pm 0.011}$ | $0.034_{\pm 0.001}$ | $0.045_{\pm 0.016}$ | $0.012_{\pm 0.001}$ | $0.028_{\pm 0.010}$ |
| Temperature | $0.230_{\pm 0.021}$ | $\mathbf{0.176}_{\pm 0.027}$ | $\mathbf{0.214}_{\pm 0.004}$ | $0.177_{\pm 0.009}$ | $0.217_{\pm 0.026}$ | $\mathbf{0.176}_{\pm 0.042}$ |
| Rain | $\mathbf{0.036}_{\pm 0.016}$ | $\mathbf{0.011}_{\pm 0.003}$ | $0.052_{\pm 0.027}$ | $0.039_{\pm 0.033}$ | $0.039_{\pm 0.024}$ | $\mathbf{0.011}_{\pm 0.004}$ |
| NN5 | $0.147_{\pm 0.008}$ | $\mathbf{0.045}_{\pm 0.007}$ | $\mathbf{0.146}_{\pm 0.015}$ | $0.080_{\pm 0.006}$ | $0.152_{\pm 0.008}$ | $0.048_{\pm 0.013}$ |
| Fred-MD | $\mathbf{0.002}_{\pm 0.001}$ | $0.201_{\pm 0.014}$ | $0.002_{\pm 0.005}$ | $\mathbf{0.199}_{\pm 0.003}$ | $\mathbf{0.002}_{\pm 0.001}$ | $0.218_{\pm 0.048}$ |
| Exchange | $\mathbf{0.137}_{\pm 0.012}$ | $1.621_{\pm 0.126}$ | $0.139_{\pm 0.011}$ | $\mathbf{1.595}_{\pm 0.023}$ | $0.139_{\pm 0.014}$ | $1.625_{\pm 0.115}$ |
| Count | **15** | | 8 | | 6 | |

similar to the original data. By observing the frequency spectrum, we can infer that our window-to-window alignment method can extract more detailed periodic patterns than TimeDP, which probably explains why our method outperforms SOTA baselines.

The experimental phenomenon is consistent with our understanding and analysis of window-to-window alignment. Since periodicity is widely presented in real-world time-series datasets, our Winformer achieves superior performance across most scenarios.

### 5.3. Ablation Study

As the ablation results in Table 2, we evaluated the performance with our **window-wise alignment**, and traditional methods, which modeling the time-series in **patch-wise alignment** and **point-wise alignment**. The window-wise alignment achieves the best performance on most of the datasets. And patch-wise alignment shows slightly advantages in dataset wind considering both KL and MMD, which is because the periodic features in this dataset is relatively weak. However, our model still has obvious superiority in most scenarios, because periodicity is widely present in

real-world time-series data. More visualized ablation results are shown in Appendix D.2.

We provide results comparing random initiate kernel and our method, with results in Fig. 7, showing our advancement in full frequency modeling. We further explore how the kernel changes during the training process in Sec. 6, showing the random initialized kernel tends to become an Avg-Pooling, which is a special case within our proposed framework.

## 6. Discussion

In this section, we further analyze the intermediate results and phenomena to understand how our method enhances the generation process.

**Q1: How does the Ample attention help the diffusion model recovering the time-series patterns?** We conduct experiments on a synthetic time-series data as Figure 5(a), and it contains sinusoidal signals with different cycle periods. The largest period is 50, and the smallest is 5. We evaluate the proposed model on the synthetic data and draw the product score's visualization between input $\mathbf{Q}$ and $\mathbf{K}$ in Eq.(1). Due to space limit, we place the score matrix of the

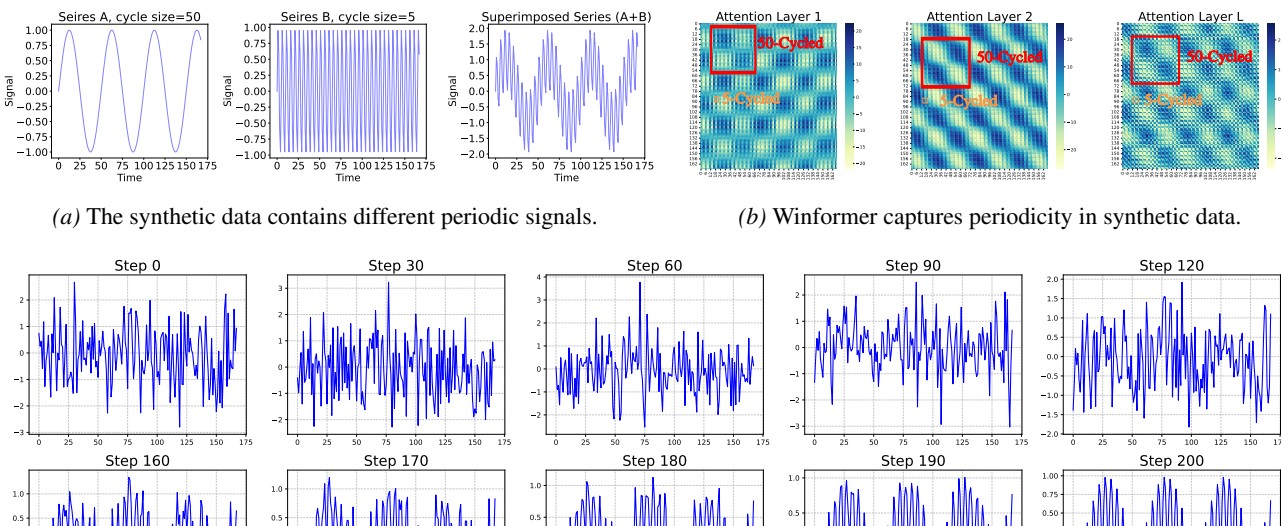

*(a)* The synthetic data contains different periodic signals.

*(b)* Winformer captures periodicity in synthetic data.

*(c)* The visualization of Winformer's denoising process on the synthetic data.

*Figure 5.* Intuitive validation on synthetic data. On the synthetic data shown in (a), the attention map produced by Winformer exhibits the pattern illustrated in (b), indicating that periodicity is one of the primary characteristics reflected in the attention map. The process of generating the final sequence is illustrated in (c), where coarse-grained long-periods (50-cycled) emerge first, followed by fine-grained short-period (5-cycled).

first two layers and the last one (more visualization figures are in Appendix D), where the darker area in heatmap indicates closer time-series interactions. The swapped color stripes reveals distinct two kinds of cycle patterns: the smaller orange grid is 5-cycled and the bigger red one is 50-cycled. Thus, we could leverage the Ample attention to capture the periodic interactions for better recovering.

**Q2: How does the Winformer architecture utilize the periodic information during the denoising process?** We visualized the denoising process in Figure 5(c), and find out that the longer cycle firstly emerges then the rest during the denoising process. Taking a concrete example at step 140, it presents a rough cycle with period 50, then the cycle period shrinks to 5 at step 180. The Winformer appears to first identify major periodic patterns that have a wider impact, then incorporate more detailed periodic patterns. This adaptive nature enables Winformer to have generalized window alignment capability in denoising process, particularly when applied to the cross-domain cases.

**Q3: How does the aligning kernel change during the training process?** To understand what the model learned within the aligning kernel, we have explored the changes of the kernel during the training process, considering on two kinds of initialization for comparison. **Considering the randomly initialized kernel weights,** the Fig. 6(a) and Fig. 6(b) represent the kernel weights before and after training with

randomly initialized weights. It can be observed that the values in the weight matrix tend to be evenly distributed, making it similar to the 'kernel' of an average pooling layer, as the weights for each position are nearly equal. As we discussed in Eq.(11), the avg(·) represents the $\psi(0)$, which is a special form of the aligning kernel. Additional results with the kernel implemented by the averaging pooling are provided in Appendix A. **If initialize with the DCT-based kernel weights,** the visualizations for kernel weights are shown in Fig. 6(c) and Fig. 6(d). It indicates that the weights change slightly, because the DCT basis shows advantages in periodicity capturing and provides sufficient guidance for denoising. Thus, the aligning kernel persists its periodicity capturing ability during training.

## 7. Conclusion

We address the challenge of periodicity alignment for cross-domain time-series generation by reforming the attention mechanism in window-wise alignment. Specifically, we propose Winformer, a diffusion framework built on window-wise modeling, which replaces pairwise similarity in vanilla attention with window-to-window comparison via time-frequency analysis. With theoretical deduction, the window-wise alignment is reducible to learnable convolutions, enabling effective implementation. Winformer shows advanced performance by achieving an average 10.67% MMD

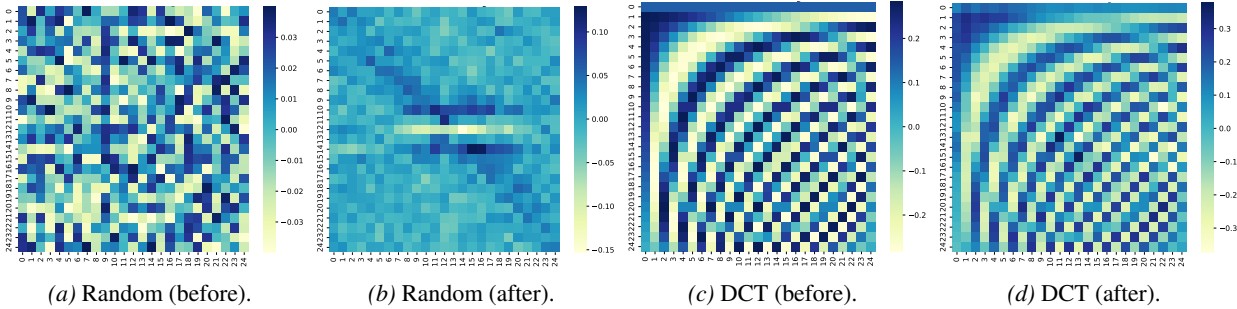

| (a) Random (before). | (b) Random (after). | (c) DCT (before). | (d) DCT (after). |

*Figure 6.* Visualization of the kernels for window-wise alignment, with random initialization or with DCT-based initialization. The visualization compares the kernel before and after training. It can be observed that kernels initialized randomly tend to converge toward average pooling after training, whereas kernels initialized with DCT largely preserve the original DCT bases after training.

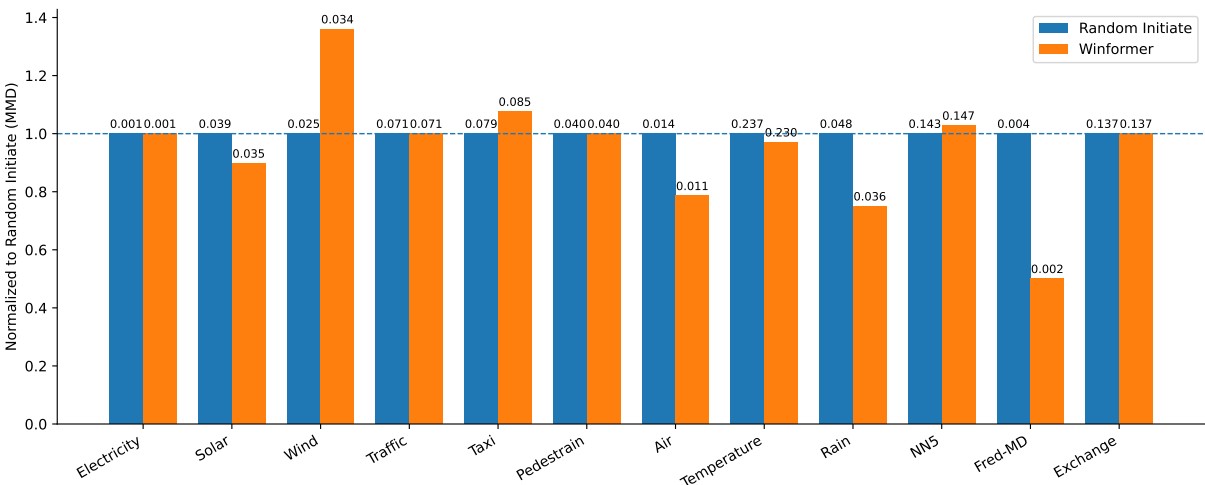

*Figure 7.* Results for ablation study on ablation study for kernel initialization. As the random initialized kernel tend to converge toward average pooling after training, which is a special case of our framework, it contains strong performance in a certain of datasets.

improvement over SOTA baselines on 12 real-world datasets. Visualization analysis and ablation studies verified the effectiveness of our window-wise alignment. Future works will focus on better conditioning, multi-modal integration, and diffusion efficiency optimization.

# 8. Limitations

Although Winformer achieves competitive time-series forecasting performance, it has limitation as its applicability boundary remains unexplored. First, the model degrades on trend-dominated data like wind and temperature data, while no usage guidance is provided for practitioners. We provide a simple idea of integrating our model with trend decomposition in Appendix F.1, showing it has the potential to further improve performance on datasets with prominent trends. Second, this work leaves the window-wise alignment extension for multivariate time-series scenarios uninvestigated. We provide a preliminary idea of combining GCN for multi-variable series in Appendix F.2. Third, it's diffi-

cult for hyper-parameter selecting for long seasonal cycles. However, the hyper-parameter $s$ can be set to a large enough value, as it could backward compatible with smaller cycles.

# Acknowledgement

This work was supported by the grants from the Natural Science Foundation of China (62572035, 62225202), and Beijing Natural Science Foundation (L248032). Thanks for the computing infrastructure provided by Beijing Advanced Innovation Center for Big Data and Brain Computing. Sponsored by CAAI-MindSpore Open Fund, developed on OpenI Community. Jianxin Li is the corresponding author.

# Impact Statement

This paper presents work whose goal is to advance the field of machine learning. There are many potential societal consequences of our work, none of which we feel must be specifically highlighted here.

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

# A. Hyper-parameter Study

In this experiment, we evaluated the performance with different kernel types and sizes. The results are shown in Table 3, and the best model, which adapts a 25-size Conv2d layer as the aligning kernel, whose results are also selected to be shown in Table 1 competing with SOTA baselines.

**For kernel types**, we apply two types of kernels. The Conv2d kernel is implemented with a 2D convolution neural network, whose kernel is initialized with the DCT basis. The Avgpool2d kernel is implemented with a 2D average pooling operator. The results show that Conv2d performs better than Avgpool2d, and both operators show effectiveness compared to the model without any kernel. We can find that DCT kernel performs better than Avg pooling because DCT kernel could capture more periodicity information, while DCT achieves 8+8+9=25 best counts and Avg achieves 4+9+5=18 best counts. Thus, in this paper, we mainly consider the DCT kernels. By the way, Avg Pooling is more efficient, so it's more suitable with limited calculation consumption. **For kernel sizes**, we select three sizes, including 7, 13 and 25, which are odd numbers to facilitate the convolution operator. The results show that, for Conv2d kernel, model with the size of 25 performs best because it contains the cycle up to 24, which is a common cycle for time-series data. The optimal 25-size kernel in the paper works because it includes typical cycles (4, 6, 12, 24) without overcomplicating computation.

*Table 3.* Results for hyper-parameters study . Best results of each line are **bold**. The best model uses 2D Convolution layer with the kernel size of 25, and it achieves best performance on 9 lines in total, which is also selected as our final model shown in Table 1.

| | # Kernel Type | Window-wise align by Conv2d kernel | | | Window-wise align by Avgpool2d kernel | | |
|---|---|---|---|---|---|---|---|
| | # Kernel Size | 7 | 13 | 25 | 7 | 13 | 25 |
| Maximum Mean Discrepancy | Electricity | $\mathbf{0.001}_{\pm 0.001}$ | $0.002_{\pm 0.003}$ | $\mathbf{0.001}_{\pm 0.002}$ | $0.002_{\pm 0.003}$ | $0.002_{\pm 0.003}$ | $0.002_{\pm 0.003}$ |
| | Solar | $0.035_{\pm 0.001}$ | $0.035_{\pm 0.002}$ | $0.035_{\pm 0.004}$ | $\mathbf{0.033}_{\pm 0.005}$ | $\mathbf{0.033}_{\pm 0.004}$ | $\mathbf{0.033}_{\pm 0.004}$ |
| | Wind | $0.035_{\pm 0.015}$ | $\mathbf{0.033}_{\pm 0.011}$ | $0.034_{\pm 0.014}$ | $\mathbf{0.033}_{\pm 0.010}$ | $\mathbf{0.033}_{\pm 0.010}$ | $0.037_{\pm 0.013}$ |
| | Traffic | $0.074_{\pm 0.007}$ | $0.074_{\pm 0.003}$ | $\mathbf{0.071}_{\pm 0.005}$ | $0.072_{\pm 0.003}$ | $\mathbf{0.071}_{\pm 0.003}$ | $0.074_{\pm 0.005}$ |
| | Taxi | $\mathbf{0.080}_{\pm 0.015}$ | $0.086_{\pm 0.012}$ | $0.085_{\pm 0.010}$ | $0.084_{\pm 0.013}$ | $0.083_{\pm 0.012}$ | $0.082_{\pm 0.011}$ |
| | Pedestrain | $\mathbf{0.040}_{\pm 0.011}$ | $0.041_{\pm 0.010}$ | $\mathbf{0.040}_{\pm 0.008}$ | $0.042_{\pm 0.009}$ | $0.041_{\pm 0.010}$ | $\mathbf{0.040}_{\pm 0.011}$ |
| | Air | $0.014_{\pm 0.004}$ | $0.013_{\pm 0.001}$ | $\mathbf{0.011}_{\pm 0.002}$ | $0.013_{\pm 0.001}$ | $0.013_{\pm 0.001}$ | $0.012_{\pm 0.002}$ |
| | Temperature | $0.226_{\pm 0.022}$ | $\mathbf{0.219}_{\pm 0.027}$ | $0.230_{\pm 0.021}$ | $0.220_{\pm 0.026}$ | $\mathbf{0.219}_{\pm 0.024}$ | $0.229_{\pm 0.023}$ |
| | Rain | $\mathbf{0.033}_{\pm 0.012}$ | $0.050_{\pm 0.046}$ | $0.036_{\pm 0.016}$ | $0.038_{\pm 0.024}$ | $0.037_{\pm 0.019}$ | $0.036_{\pm 0.016}$ |
| | NN5 | $0.149_{\pm 0.005}$ | $0.151_{\pm 0.007}$ | $\mathbf{0.147}_{\pm 0.008}$ | $0.153_{\pm 0.006}$ | $0.154_{\pm 0.006}$ | $0.154_{\pm 0.007}$ |
| | Fred-MD | $\mathbf{0.002}_{\pm 0.001}$ | $\mathbf{0.002}_{\pm 0.001}$ | $\mathbf{0.002}_{\pm 0.001}$ | $\mathbf{0.002}_{\pm 0.001}$ | $\mathbf{0.002}_{\pm 0.001}$ | $\mathbf{0.002}_{\pm 0.001}$ |
| | Exchange | $\mathbf{0.136}_{\pm 0.012}$ | $0.139_{\pm 0.015}$ | $0.137_{\pm 0.012}$ | $0.140_{\pm 0.015}$ | $0.139_{\pm 0.014}$ | $0.138_{\pm 0.014}$ |
| K-L Divergence | Electricity | $0.019_{\pm 0.030}$ | $0.020_{\pm 0.018}$ | $\mathbf{0.008}_{\pm 0.010}$ | $0.033_{\pm 0.044}$ | $0.023_{\pm 0.021}$ | $0.014_{\pm 0.010}$ |
| | Solar | $0.021_{\pm 0.013}$ | $\mathbf{0.012}_{\pm 0.004}$ | $0.013_{\pm 0.005}$ | $0.015_{\pm 0.008}$ | $0.015_{\pm 0.009}$ | $0.016_{\pm 0.012}$ |
| | Wind | $0.201_{\pm 0.040}$ | $0.186_{\pm 0.030}$ | $0.202_{\pm 0.044}$ | $\mathbf{0.180}_{\pm 0.026}$ | $0.182_{\pm 0.026}$ | $0.203_{\pm 0.040}$ |
| | Traffic | $0.013_{\pm 0.007}$ | $\mathbf{0.009}_{\pm 0.003}$ | $0.011_{\pm 0.002}$ | $0.010_{\pm 0.003}$ | $\mathbf{0.009}_{\pm 0.002}$ | $0.011_{\pm 0.005}$ |
| | Taxi | $0.006_{\pm 0.004}$ | $\mathbf{0.004}_{\pm 0.003}$ | $0.005_{\pm 0.003}$ | $0.005_{\pm 0.002}$ | $\mathbf{0.004}_{\pm 0.001}$ | $\mathbf{0.004}_{\pm 0.001}$ |
| | Pedestrain | $\mathbf{0.009}_{\pm 0.004}$ | $0.011_{\pm 0.005}$ | $\mathbf{0.009}_{\pm 0.004}$ | $0.012_{\pm 0.004}$ | $0.012_{\pm 0.004}$ | $0.010_{\pm 0.004}$ |
| | Air | $0.031_{\pm 0.012}$ | $0.025_{\pm 0.009}$ | $0.026_{\pm 0.011}$ | $0.028_{\pm 0.010}$ | $0.027_{\pm 0.009}$ | $\mathbf{0.023}_{\pm 0.008}$ |
| | Temperature | $\mathbf{0.173}_{\pm 0.027}$ | $0.174_{\pm 0.039}$ | $0.176_{\pm 0.027}$ | $0.178_{\pm 0.036}$ | $0.181_{\pm 0.031}$ | $0.186_{\pm 0.030}$ |
| | Rain | $0.014_{\pm 0.008}$ | $\mathbf{0.009}_{\pm 0.004}$ | $0.011_{\pm 0.003}$ | $0.011_{\pm 0.004}$ | $0.015_{\pm 0.008}$ | $0.012_{\pm 0.004}$ |
| | NN5 | $0.048_{\pm 0.009}$ | $0.049_{\pm 0.018}$ | $\mathbf{0.045}_{\pm 0.007}$ | $0.047_{\pm 0.012}$ | $0.054_{\pm 0.018}$ | $0.053_{\pm 0.026}$ |
| | Fred-MD | $0.248_{\pm 0.109}$ | $\mathbf{0.197}_{\pm 0.028}$ | $0.201_{\pm 0.014}$ | $0.201_{\pm 0.020}$ | $\mathbf{0.197}_{\pm 0.020}$ | $0.204_{\pm 0.024}$ |
| | Exchange | $1.629_{\pm 0.166}$ | $1.594_{\pm 0.151}$ | $1.621_{\pm 0.126}$ | $1.610_{\pm 0.159}$ | $\mathbf{1.590}_{\pm 0.155}$ | $1.635_{\pm 0.125}$ |
| | Count | 8 | 8 | **9** | 4 | 9 | 5 |

# B. More Ablations

The ablation results for window aligning and domain prompting is shown in Table 4, indicating the necessity of these parts. Additionally, our prompting is different from TimeDP. TimeDP injects prompts via attention masking, while we use adaptive layer with multi-cond instead, avoiding interference with periodicity extraction in attention.

*Table 4.* Results for ablation study on different parts, including window aligning and domain prompting. The best results are **blod**.

| Ablation Type | Winformer | W.O. window aligning | W.O. domain prompt | W.O. both |
|---|---|---|---|---|
| Electricity | $0.001_{\pm 0.002}$ | $0.002_{\pm 0.004}$ | $0.009_{\pm 0.006}$ | $0.009_{\pm 0.007}$ |
| **Solar** | $0.035_{\pm 0.004}$ | $0.035_{\pm 0.003}$ | $0.402_{\pm 0.077}$ | $0.419_{\pm 0.073}$ |
| Wind | $0.034_{\pm 0.014}$ | $0.034_{\pm 0.010}$ | $0.121_{\pm 0.088}$ | $0.115_{\pm 0.074}$ |
| Traffic | $0.071_{\pm 0.005}$ | $0.073_{\pm 0.008}$ | $0.212_{\pm 0.037}$ | $0.221_{\pm 0.035}$ |
| Taxi | $0.085_{\pm 0.010}$ | $0.088_{\pm 0.011}$ | $0.101_{\pm 0.031}$ | $0.127_{\pm 0.030}$ |
| Pedestrain | $0.040_{\pm 0.008}$ | $0.041_{\pm 0.010}$ | $0.077_{\pm 0.012}$ | $0.057_{\pm 0.011}$ |
| Air | $0.011_{\pm 0.002}$ | $0.012_{\pm 0.001}$ | $0.030_{\pm 0.032}$ | $0.031_{\pm 0.022}$ |
| temperature | $0.230_{\pm 0.021}$ | $0.217_{\pm 0.026}$ | $0.60_{\pm 0.320}$ | $0.58_{\pm 0.330}$ |
| rain | $0.036_{\pm 0.016}$ | $0.039_{\pm 0.024}$ | $0.192_{\pm 0.070}$ | $0.189_{\pm 0.058}$ |
| NN5 | $0.147_{\pm 0.008}$ | $0.152_{\pm 0.008}$ | $0.215_{\pm 0.023}$ | $0.259_{\pm 0.033}$ |
| Fred-MD | $0.002_{\pm 0.001}$ | $0.002_{\pm 0.001}$ | $0.004_{\pm 0.001}$ | $0.004_{\pm 0.001}$ |
| Exchange | $0.137_{\pm 0.012}$ | $0.139_{\pm 0.014}$ | $0.140_{\pm 0.044}$ | $0.138_{\pm 0.046}$ |

# C. Computation Costs

In this section, we try to analyze the computational costs of our method.

Theoretically, we assume the length of time-series if $L_x$, and the dimension is $D$. The computation complexity of the calculation for similarity scores, which is the dot-product of $\mathbf{Q}$ and $\mathbf{K}$, is $\mathcal{O}(L_x^2 D)$. If the windows size is set to $s$, which means to compute with convolutional network with kernel size $s$, the computational complexity is $\mathcal{O}(s^2)$. We travel through all the time points of $\mathbf{Q}$ and $\mathbf{K}$, the total computation of calculating $\mathbf{S}$ becomes $\mathcal{O}(s^2 \cdot L_x^2 D)$, where the original similarity calculation cost is $\mathcal{O}(L_x^2 D)$.

We also tested the actual costs of Winformer as shown in Table 5. Notably, the cost time is for the whole pipeline, which is not limited to the ample attention. The training speed is to test how fast a denoising model can denoising a batch of data with gradient backpropagation during the training process. The sampling speed evaluates how fast a denoising model can make a prediction during the testing process. Although our window-wise alignment increases the complexity, in practical experiments, the actual computation consumption is acceptable with slight increase comparing with point-wise alignment because the calculation of convolutional neural operators has been optimized by machine learning frameworks.

*Table 5.* Computational costs for different kinds of alignment method. We evaluate the average speed of training and sampling of each batch (batch size=128).

| Model | Kernel Size | Train Speed (batch/s) | Sample Speed (batch/s) |
|---|---|---|---|
| Point-wise Alignment | - | 2.92 | 10.57 |
| Winformer (Window-wise Alignment) | 7 | 2.58 | 9.69 |
| | 13 | 2.33 | 9.13 |
| | 25 | 2.30 | 8.95 |

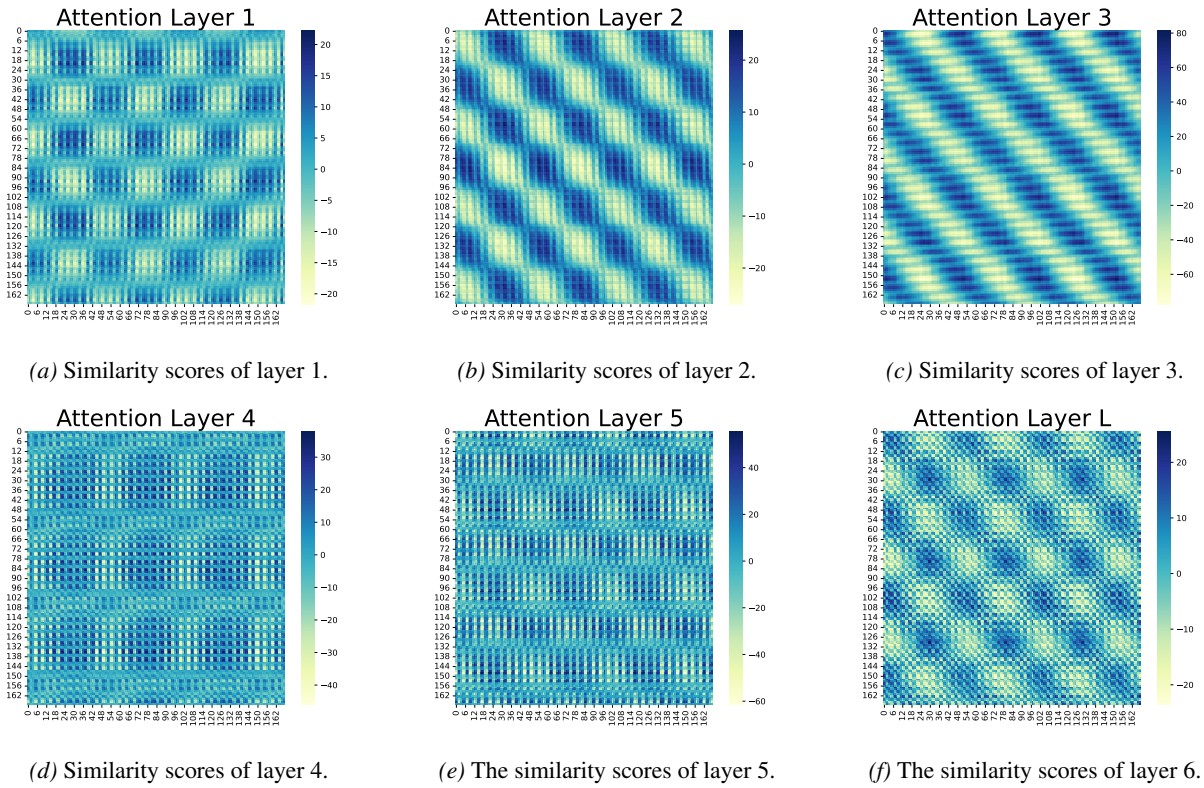

*(a)* Similarity scores of layer 1.    *(b)* Similarity scores of layer 2.    *(c)* Similarity scores of layer 3.

*(d)* Similarity scores of layer 4.    *(e)* The similarity scores of layer 5.    *(f)* The similarity scores of layer 6.

*Figure 8.* Visualization of the similarity matrices for synthetic data. On the synthetic data, the attention map for multiple layers indicates that periodicity is one of the primary characteristics reflected in the attention map.

## D. Visualizations and Discussions

We provide extended visualized figures on real-world datasets, including the visualization of the attention similarity matrix, the visualization of the denoising process and the visualization of the generated time-series. We further discuss about these visualized figures in the following subsections.

### D.1. Visualization of the Attention

In this subsection, we want to explore how can the Ample attention help the diffusion model in recovering the time-series patterns. We conduct experiments on the synthetic datasets to discover whether the model utilize the periodic pattern in denoising, which inspire us to enhance the periodic features with the Ample attention. We also verify the effectiveness on periodic enhancement by visualizing the similarity score before and after window-wise alignment.

#### D.1.1. VISUALIZATION ON SYNTHETIC DATA

We conduct the experiments on the synthetic datasets, which contains various periodic patterns as defined in Eq. 19. By observing the phenomena on the synthetic dataset, we can explore how the denoising model works with periodic capturing. We visualized the similarity score of the attention mechanism in Figure 8. In the heatmap of the score, the darker the color represents the deeper similarity of the time-series points. These visualized images show periodic cycles with repetitive square-like patterns. From layer 1 to Layer $L$ ($L = 6$), we can find that the patterns may shift with constant steps. The phenomena enlighten us that the periodic features are essential for transformer-based time-series denoising models.

#### D.1.2. VISUALIZATION ON REAL-WORLD DATA

We also visualized the similarity score matrices of the self-attention mechanism during the model's denoising process on a portion of real-world data, as shown in Figure 9. This figure corresponds to the taxi dataset, which contains strong periodic patterns. The three subgraphs represent the kernel for window-wise alignment, the similarity score matrix before alignment,

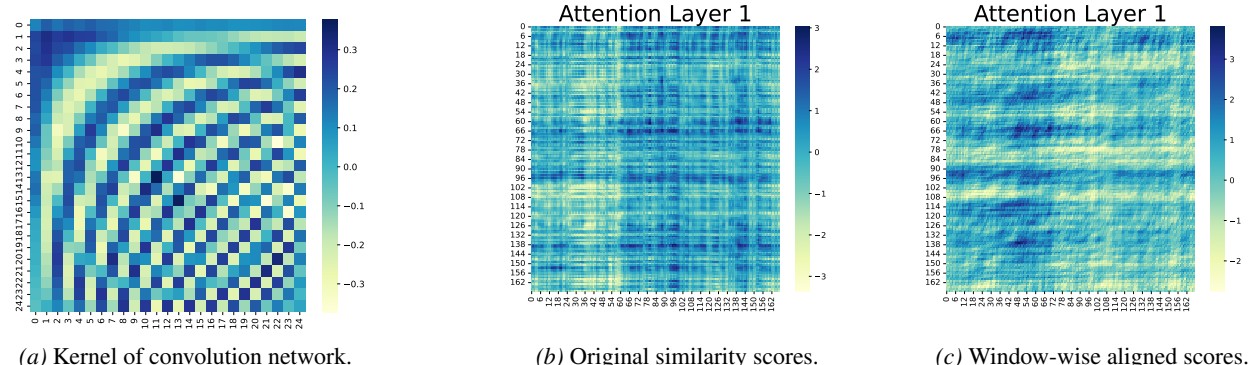

*(a)* Kernel of convolution network.      *(b)* Original similarity scores.      *(c)* Window-wise aligned scores.

*Figure 9.* Visualization of the kernel for window-wise alignment, the similarity matrix before alignment, and the similarity matrix after window-wise alignment for dataset Taxi.

and the similarity matrix after window-wise alignment. The time-series data in the selected real-world datasets exhibit good periodicity, so their similarity matrices show a grid pattern. After window-wise alignment, we notice that the long-range grids become more distinct, indicating the window-wise alignment fuse features inside the observation window. Thus, our method can enhance the periodic features.

### D.2. Visualization for alignment comparison

To explore the differences in the effectiveness of the three kinds alignments, namely point-wise, patch-wise and window-wise, in extracting complex periodicity, we conducted result visualization on taxi dataset. Taxi dataset contains complex periodicity structure, as not only presents the daily cycle but also includes the detailed periodic changes of morning and evening peaks. We visualized the series generated with point-wise, patch-wise and window-wise alignment in Fig. 10. We can find that, window-wise method could capture the periodicity better than other methods, by well establishing periodicity structures.

### D.3. Visualization of the Generated Time-series

We visualized some time-series data generated by the Winformer, as shown in Fig. 11. The solar dataset contains a simple daily cycle. These generated series, containing obvious periodicity, indicates that the Winformer can effectively capture the periodic features. This confirms that our window-wise alignment can enhance the capturing of periodicity.

### D.4. Visualization of the Denoising Process

To further explore the denoising process in time-series generation tasks, we visualized the sequences during the denoising process on real-world dataset, shown in Figure 12. The Solar datasets contains obvious daily cycles. By observing these images, we can find that in the denoising process of time-series data, long-range periodic information is captured first and displayed initially. Other detailed information will be revealed in later steps of denoising. Therefore, we propose the hypothesis that in the view of denoising model, the basic components for time-series data are frequency features instead of temporal points. That's explain and verify why the Winformer can achieve better performance against SOTA baselines.

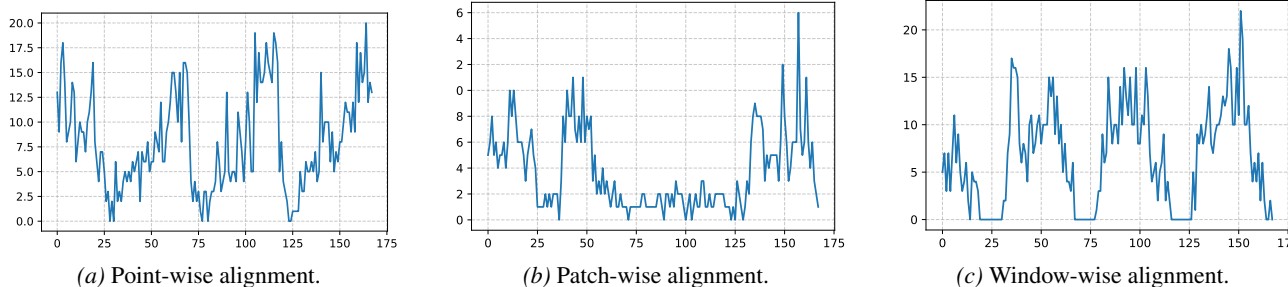

*(a)* Point-wise alignment.    *(b)* Patch-wise alignment.    *(c)* Window-wise alignment.

*Figure 10.* Visualization of the series generated by point-wise, patch-wise and window-wise alignment for dataset Taxi. This example demonstrates that window-wise generated samples capture periodicity more effectively.

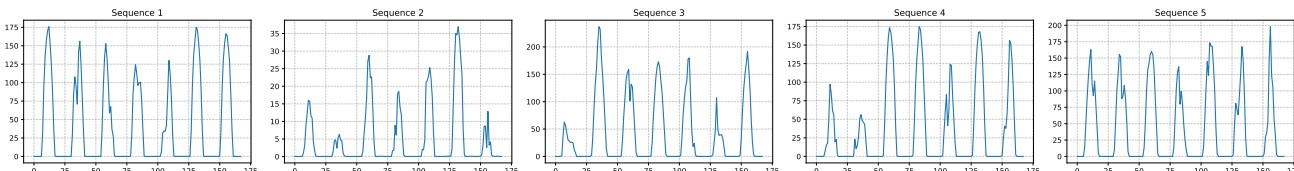

*Figure 11.* The generated time-series data for dataset Solar, containing obvious periodicity.

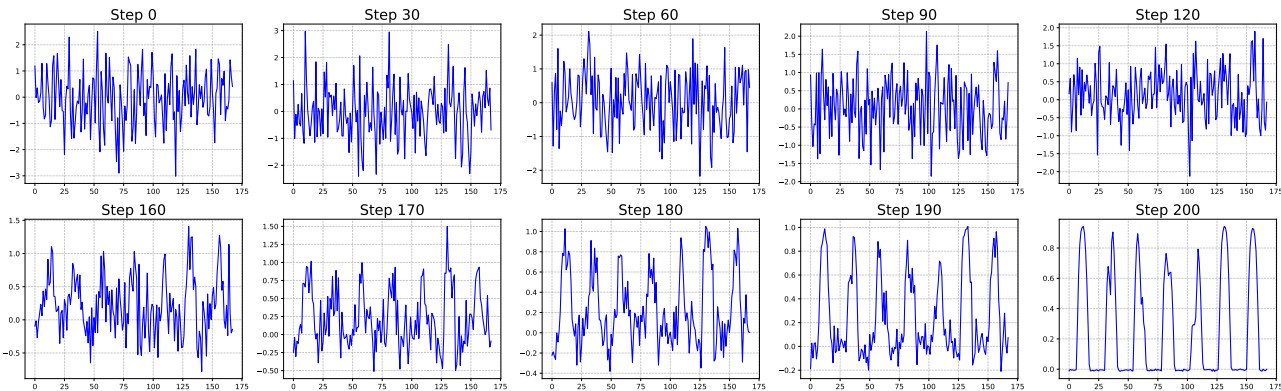

*Figure 12.* Denoising Process for the dataset of Solar, showing how periodicity reveals throughout the denoising process.

# E. Domain Discrepancy

We add a supplementary evaluation on domain discrepancy. Specifically, we evaluate the KL-divergence between different domains. The results are shown in Fig 13. To facilitate observation, all data in the figure are logarithmically transformed (log-transformed). For most datasets, the KL divergence between the generated data and the real data in their respective domains is the lowest, indicating that the domain-specific features are well learned.

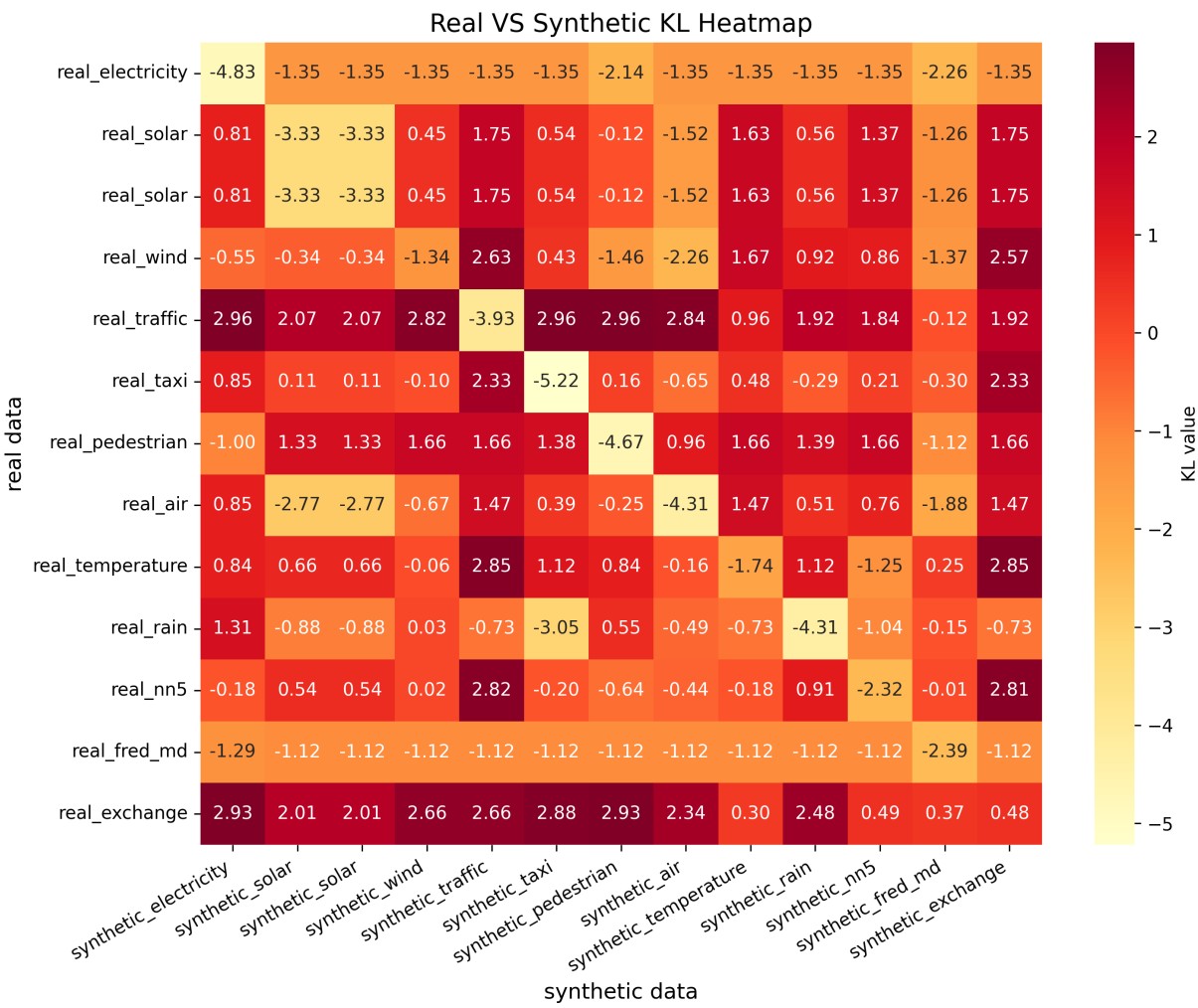

*Figure 13.* The domain discrepancy among datasets. For most datasets, domain discrepancy is evident, indicating that the domain-specific features are well learned.

# F. Expansion form

### F.1. Integrating with Trend decomposition.

We integrate Winformer with a basic trend adaptation (Autoformer-style) in Table 6, showing it has the potential to further improve performance on datasets with prominent trends.

### F.2. Multivariate Time-series Generation

Our model outperforms baseline on PEMS datasets, which are multivariate time-series datasets with complex cross-channel dependencies, with results in Table 7. We just inject the cross-domain dependencies (captured by a GCN) as condition to our model, and our model can realize the channel jointly generation with the dependencies.

*Table 6.* Integrating Winformer with Trend decomposition.

| Alignment Type | Winformer | | Winformer+Trend | |
|---|---|---|---|---|
| Metric | MMD | K-L | MMD | K-L |
| Wind | $0.034_{\pm 0.014}$ | $0.202_{\pm 0.044}$ | $\mathbf{0.030}_{\pm 0.009}$ | $\mathbf{0.200}_{\pm 0.041}$ |
| Fred-MD | $\mathbf{0.002}_{\pm 0.001}$ | $0.201_{\pm 0.014}$ | $\mathbf{0.002}_{\pm 0.001}$ | $\mathbf{0.199}_{\pm 0.017}$ |
| Temperature | $0.230_{\pm 0.021}$ | $0.176_{\pm 0.027}$ | $\mathbf{0.202}_{\pm 0.025}$ | $\mathbf{0.170}_{\pm 0.028}$ |

*Table 7.* Results on PEMS datasets (cross-channel datasets).

| Model | TimeDP | | Winformer | |
|---|---|---|---|---|
| Metric | KL | MMD | KL | MMD |
| PEMS03 | 1.67 | 0.34 | **1.53** | **0.29** |
| PEMS04 | 2.29 | 0.33 | **1.88** | **0.21** |
| PEMS08 | 1.80 | 0.34 | **1.61** | **0.28** |

## F.3. Downstream Evaluation

Additional results show our practical value in Table 8 and 9, for classification and forecasting as you recommended. The predictor and classifier are vanilla transformer, and our method can help them to achieve better or comparable performance.

*Table 8.* Results on Application (multivariate forecasting).

| Model | TSTR | | Original | |
|---|---|---|---|---|
| Metric | MSE | MAE | MSE | MAE |
| ETTh1 | 0.7411 | 0.6727 | 0.7682 | 0.6898 |
| ETTh2 | 1.3013 | 0.9295 | 1.3159 | 0.9352 |
| ETTm1 | 0.4974 | 0.5013 | 0.4997 | 0.502 |
| ETTm2 | 0.3881 | 0.4851 | 0.3785 | 0.4742 |
| ECL | 0.2435 | 0.3432 | 0.2552 | 0.3481 |

## F.4. On Medical Datasets

To directly address the concern on irregular medical data, we now evaluate on three additional medical datasets. As shown in Table 10, Winformer consistently outperforms both TimeDP and DiT across all three datasets on both metrics.

*Table 9.* Results on Application (Classification).

| Model | Add sythetic data | Original |
|---|---|---|
| EthanolConcentration | 0.2776 | 0.2738 |
| FaceDetection | 0.6827 | 0.6765 |
| Handwriting | 0.3835 | 0.3753 |
| Heartbeat | 0.7756 | 0.7707 |
| JapaneseVowels | 0.9757 | 0.9784 |
| PEMS-SF | 0.8382 | 0.8324 |
| SpokenArabicDigits | 0.9859 | 0.9845 |
| UWaveGestureLibrary | 0.8625 | 0.8594 |

*Table 10.* Evaluation on Medical Data.

| metric | Winformer(Ours) | | TimeDP | | DiT | |
|---|---|---|---|---|---|---|
| | KL | MMD | KL | MMD | KL | MMD |
| PTB-XL | 1.424 | 0.3496 | 1.552 | 0.3532 | 1.446 | 0.3512 |
| illness | 1.628 | 0.5979 | 1.6455 | 0.6708 | 1.637 | 0.6262 |
| Health | 1.502 | 0.4174 | 1.685 | 0.4377 | 1.613 | 0.4283 |

# G. Experiment Details

We further report the detailed settings of the experiments, including information of the measurement, datasets and implementation.

## G.1. Metrics and Measurement

In this section, we describe the metrics and other settings which related to the measurement of the model's performance. First, we formulate the key metrics evaluating the performance of the time-series generation. Then we describe the detail settings for repeated experiments and how we get the reported results with error bounds.

### G.1.1. FORMULATED METRICS

In this paper, we adopt two metrics in measuring the quality of the generated time-series data. Firstly, we define a real time-series data with $L$ length and $D$ channels as $\mathbf{X} = [\mathbf{X}_1, \mathbf{X}_2, ..., \mathbf{X}_L] \in \mathbb{R}^{L \times D}$, and the synthetic data is $\hat{\mathbf{X}} = [\hat{\mathbf{X}}_1, \hat{\mathbf{X}}_2, ..., \hat{\mathbf{X}}_L] \in \mathbb{R}^{L \times D}$. Then we formulate the metrics of Maximum Mean Discrepancy and Kullback-Leibler Divergence as follow.

**(1) Maximum Mean Discrepancy (MMD).** The MMD is a distribution similarity evaluation method. Specifically, we transform the time-series data into a high-dimension space by $\Phi(\cdot, \cdot)$. Then we calculates the average of the results obtained by the kernel to get the MMD, which is formulated as follow:

$$L_{\text{MMD}} = \frac{\sum_{i=1}^{N} \Phi(\mathbf{X}_i, \mathbf{X}_i)}{N} + \frac{\sum_{i=1}^{N} \Phi(\hat{\mathbf{X}}_i, \hat{\mathbf{X}}_i)}{N} - 2 \frac{\sum_{i=1}^{N} \Phi(\mathbf{X}_i, \hat{\mathbf{X}}_i)}{N} \quad , \tag{15}$$

where $\Phi(\cdot, \cdot)$ is implemented by the radial basis function kernel.

**(2) Kullback-Leibler Divergence(K-L).** The K-L is a common metric measuring the similarity between real and synthetic data.

$$L_{\text{K-L}} = \sum_{i=1}^{K} P(\mathbf{X}) \log \left( \frac{P(\mathbf{X})}{Q(\hat{\mathbf{X}})} \right) \quad , \tag{16}$$

where $P(\cdot)$ and $Q(\cdot)$ are mapping functions to obtain the distribution of the data by reforming the data into histogram, which includes $K$ indexes in total.

### G.1.2. REPEATED EXPERIMENTS

All experiments are repeated five times with random seeds ranging from 2021 to 2025. To demonstrate the comprehensive effect of the model, we report the average results with their standard deviation of the repeated experiments.

## G.2. Datasets

We conducted the time-series generation experiment on 12 real-world datasets and a synthetic dataset. The description of the datasets are as follow.

### G.2.1. REAL-WORLD DATASETS

In this paper, we conduct the experiments following the setting of the TimeDP (Huang et al., 2025). The experiments contain 12 real-world datasets from four domains, including energy, economic, weather and transportation. The pre-processed datasets are open-sourced by TimeDP[1]. We list the details of the datasets in Table 11.

### G.2.2. SYNTHETIC DATASET

The synthetic datasets contains time-series data which are the combination of the sinusoidal signal with different periods. Specifically, let $\mathbf{X}_1(t)$ be the sinusoidal signal with a cycle of 50, which is defined as follow:

$$\mathbf{X}_1(t) = sin(\frac{2\pi}{50}t) \quad , \tag{17}$$

---

[1]https://huggingface.co/datasets/YukhoW/TimeDP-Data/blob/main/TimeDP-Data.zip

*Table 11.* Details for the real-world datasets.

| Domain | Dataset | Variables | Sampling interval | Description | Source |
|---|---|---|---|---|---|
| Energy | Electricity | 321 | 1 hour | the electricity consumption | UCI |
| | Solar | 137 | 1 hour | the solar power production | State of Alabama |
| | Wind | 1 | 4 second | the wind power production | AEMO |
| Weather | Air | 270 | 1 hour | the air quality levels | KDDCup2018 |
| | Temperature | 422 | 1 day | the temperature observations | Australia |
| | Rain | 422 | 1 day | the rain forecast | Australia |
| Transportation | Traffic | 963 | 1 hour | the occupancy rate of car lanes | San Francisco bay |
| | Taxi | 1214 | 30 minutes | the taxi rides | New York |
| | Pedestrian | 1 | 1 hour | the pedestrian counts | Melbourne city |
| Economic | NN5 | 111 | 1 day | the cash withdrawals from ATMs | UK |
| | Fred-MD | 107 | 1 month | the macro-economic indicators | Federal Reserve Bank |
| | Exchange | 8 | 1 day | the exchange rate | Reports of 8 countries |

and $\mathbf{X}_2(t)$ be the sinusoidal signal with a cycle of 5 defined as:

$$\mathbf{X}_2(t) = sin(\frac{2\pi}{5}t) \quad . \tag{18}$$

Then, we can obtain the synthetic data by superposing these two series as

$$\mathbf{X}(t) = \mathbf{X}_1(t) + \mathbf{X}_2(t) \quad . \tag{19}$$

Thus, we get the synthetic data $\mathbf{X}(t)$ which contains two types of cycles. We sample 20000 time steps and split the sequences into 168-length time series data. By conducting denoising method on the synthetic data, we can explore how the Ample attention and the Winformer helps in frequency capturing and cross-domain time-series generation. From visualization of the denoising process, we can find that the model firstly recovers the large-range periodic patterns. This motivates us to enhance the periodic pattern capturing by window-wise alignment, which is exactly our method does.

### G.3. Implementation Settings

In this section, we report the detailed implementation, including hyper-parameters and environments.

#### G.3.1. HYPER PARAMETERS

Our method, namely Winformer, is a transformer-based denoising model, consisting of several encoder blocks and a conditioning block. We have listed the detailed hyper-parameters of each component, which are shown in Table 12.

#### G.3.2. HARD-WARES AND ENVIRONMENTS

The experiments are conducted with an Ascend 910b GPU, with 32GB memory. Our model relies on public environment libraries, including Python, MindSpore, etc. The framework is based on the source code of TimeDP[2]. Our codes are reported in the supplementary material. More detailed environment and installation tips are reported in the ReadMe file in codes folder.

---

[2]https://github.com/microsoft/TimeCraft/tree/main/TimeDP

*Table 12.* Hyper-parameters for the Winformer.

| Catergory | Module | Name | Value |
|---|---|---|---|
| | | Dimension | 32 |
| | Conditioning block | Channel | 1 |
| | | Latent | 1 |
| | | Channel | 1 |
| | | Hidden size | 512 |
| | | Vanilla attention heads | 8 |
| Architecture | Hybrid encoder block | Ample attention heads | 8 |
| of Winformer | | MLP ratio | 4.0 |
| | | Kernel type | Conv2d |
| | | Kernel initialization | discrete cosine transform (DCT) |
| | | Kernel size | 25 |
| | | Hybrid encoder block | 6 |
| | DiT encoder block | Channel | 1 |
| | | Vanilla attention heads | 16 |
| | | MLP ratio | 4.0 |
| Forward/Reverse Process | | Noising/Denoising steps | 200 |
| | | Loss type | L1 loss |
| | | Batch size | 128 |
| Trainer | | Learning rate | 5e-5 |
| | | Train steps | 50, 000 |

# H. Task Introduction

## H.1. Generation VS. Forecasting

Forecasting tasks emphasize on detailed perception for instances, while generation tasks requires global structure awareness for distributions. Forecasting and imputation tasks focus on estimating the series to the groundtruth under know observations in every single instances, but generation is to estimate the global distribution to the whole dataset. Generation tasks impose stronger requirements on more widely existing structures while downplaying the unique characteristics of individual samples. That's why we select the generation tasks to explore how could the model learn the periodicity structure better.

## H.2. Observations for Cross-domain Generation

Cross-domain generation requires models to learn and transfer periodic patterns across diverse domains. As illustrated in Fig. 14(a), series from domain 1 and domain 2 show different periodicity, which is challenging for existing point-wise and patch-wise methods. Capturing the domain-specific periodicity directly tests the architecture's ability to generalize periodicity.

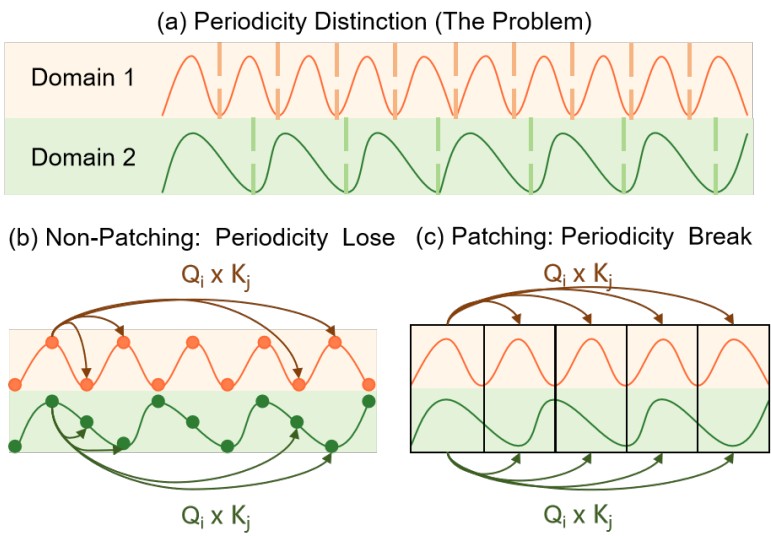

*Figure 14.* The observation for cross-domain time-series generation. For non-patching (point-wise) methods, the model understands each point relatively independently and lacks the ability to capture the correlation between adjacent time steps, which is detrimental to the acquisition of pattern structures. For patching methods, with information isolated and compressed at equal intervals, fixed windows will disrupt the periods of some sequences.

## H.3. The Discovery of the Adaptive Window-Wise Architecture

The design of Winformer's adaptive window-wise architecture stems from a key observation, that is the limitations of existing point-wise and patch-wise paradigms. Traditional point-wise attention struggles with complex dependencies and fails to model cyclic structures, as it only captures point-to-point interactions. As illustrated in Fig. 14(b), it's difficult to identify repetitive structures when time-series are numerically sensitive in continues space. Patch-wise methods fragment time-series into fixed segments, breaking evolving periodic patterns and requiring additional distribution restoration steps . Especially for cross-domain generation, models may process time-series with different windows, patching with a fixed segments could not be suitable for every domain, as illustrated in Fig. 14(c) . These flaws motivated us to rethink the "processing unit" of attention: Could we shift the discrete points or patches to continuous windows that naturally encapsulate periodic cycles? Compared to the point-wise methods, the window-wise methods can summary local features and simplified attention's work. Compared to the patch-wise mwthods, the window alignment is sliding, which allows to adapt different domains, as for every domain there is expected to have partial of the windows can contain the proper periodicity features. With these observations, we start for theoretical analysis to explore how to capture the periodicity with proper approaches.

# I. Theoretical Explanation and Proof

In this section, we supplement theoretical backgrounds to facilitate presentation of the motivation and the design concepts. With these further explanation, we can better understand the underlying reasons and deduction process.

## I.1. The Fourier Transform

The Fourier Transform (Duhamel & Vetterli, 1990) is a mathematical operator that converts signal $\mathbf{x}$ to its frequency domain representation $\widetilde{\mathbf{x}}$. It is defined as a function $\mathcal{F}$ as:

$$\mathcal{F}\{u\}(\xi) = \int_{-\infty}^{+\infty} u(x)e^{-2\pi jx\xi}dx \quad . \tag{20}$$

The $u(x)$ is the input function to signal $\mathbf{x}$'s formulation. Then, we set the integral results as individual spectral components for different frequency $\xi$.

## I.2. Calculating Similarity Score in Frequency Domain

The attention mechanism, which calculates as follow:

$$\mathcal{A} = \text{Softmax}(\frac{\mathbf{Q}\mathbf{K}^{\mathrm{T}}}{\sqrt{d}})\mathbf{V} \quad , \tag{21}$$

where $\mathbf{Q}$ and $\mathbf{K}$ are mapped features from input $x$. To calculate the similarity score $\mathbf{S} = \mathbf{Q}\mathbf{K}^{\mathrm{T}}$ in the frequency domain, we can directly transform the input data with Fourier transform and calculate the score using the Fourier components. However, such a process is time-consuming and inflexible and with Fourier operators. For a learning-based model, we prefer a learning surrogator for this operation, which can be adaptively parameterized cooperating with the model. Thus, we need to further explore the simplified calculation of the score $\mathbf{S}$.

## I.3. Convolution Theorem

To further explore the simplified calculation of the score, we have to consider how to transform the dot-product operator in frequency domain. As the convolution theorem (McGillem & Cooper, 1991) sates that:

$$\mathcal{F}\{u * v\} = \mathcal{F}\{u\} \cdot \mathcal{F}\{v\} \quad , \tag{22}$$

where $*$ is the convolution and the operator $\cdot$ represents the point-wise multiplication. The above derivations also apply for the discrete Fourier transform (DFT). Thus, there is the possibility of transforming the dot-production in the frequency domain into the convolution in the temporal domain. As a result, we can directly calculate the score $\mathbf{S}$ with convolution network, in the place of a series of the calculation for frequency transformation.

## I.4. Parseval's Theorem

For discrete time signals, we consider a time-series data $x$ with length $n$. The Parseval's Theorem states:

$$\sum_{n=0}^{N-1} |x[n]|^2 = \frac{1}{N} \sum_{k=0}^{N-1} |X[k]|^2, \tag{23}$$

where $X[k]$ is the discrete Fourier transform (DFT) of $x[n]$. The Parseval's Theorem enables us to transfer correlation of frequency domain into that to time domain. Thus, we can adapts the deduction of Eq. 11 in the attention mechanism, leading to the final format of the Ample attention shown in Eq. 12.

## I.5. Relationship Between DCT and DFT Relationship

**Lemma I.1.** *Let $x[n] \in \mathbb{R}$ be a 1D real-valued finite-length signal of length $N$ ($n = 0, 1, \ldots, N-1$). The N-point Type II Discrete Cosine Transform (DCT-II) of $x[n]$ is a **special form** of the Discrete Fourier Transform (DFT), which can be exactly derived from the DFT of the even-symmetric extended version of $x[n]$.*

*Proof.* Firstly, we define the notations for DCT and DFT, as well as the periodic even-symmetric extention of the series $x[n]$:

- $N$-point 1D DFT: $X[k] = \text{1D-DFT}\{x[n]\} = \sum_{n=0}^{N-1} x[n]e^{-j\frac{2\pi}{N}kn}$, $k = 0, 1, \ldots, N-1$, $j = \sqrt{-1}$;

- $N$-point 1D DCT-II: $X_C[k] = \text{1D-DCT-II}\{x[n]\} = \sum_{n=0}^{N-1} x[n]\cos\left(\frac{\pi}{N}\left(k + \frac{1}{2}\right)n\right)$, $k = 0, 1, \ldots, N-1$;

- 4N-point periodic even-symmetric extension of $x[n]$: $x_{\text{sym}}[n]$, satisfying $x_{\text{sym}}[n] = x_{\text{sym}}[4N - n]$ for all $n$.

By $\cos\theta = \frac{e^{j\theta} + e^{-j\theta}}{2}$, the cosine kernel is rewritten as a combination of complex exponentials, which is the core kernel of DFT:

$$\cos\left(\frac{\pi}{N}\left(k + \frac{1}{2}\right)n\right) = \frac{1}{2}\left[e^{j\frac{\pi(2k+1)n}{2N}} + e^{-j\frac{\pi(2k+1)n}{2N}}\right]. \tag{24}$$

Substituting into the DCT-II definition gives:

$$X_C[k] = \frac{1}{2}\sum_{n=0}^{N-1} x[n]\left[e^{j\frac{\pi(2k+1)n}{2N}} + e^{-j\frac{\pi(2k+1)n}{2N}}\right]. \tag{25}$$

Then, the 4N-point even-symmetric extension of the signal is defined as:

$$x_{\text{sym}}[n] = \begin{cases} x[n], & 0 \le n \le N-1, \\ x[2N - 1 - n], & N \le n \le 2N - 1, \\ 0, & 2N \le n \le 4N - 1. \end{cases} \tag{26}$$

The DFT of $x_{\text{sym}}[n]$ is $X_{\text{sym}}[m] = \sum_{n=0}^{2N-1} x_{\text{sym}}[n]e^{-j\frac{\pi mn}{2N}}$ for $m = 0, 1, \ldots, 4N - 1$. Splitting the summation and substituting $n' = 2N - 1 - n$ for the second term yields:

$$X_{\text{sym}}[m] = \sum_{n=0}^{N-1} x[n]\left[e^{-j\frac{\pi mn}{2N}} + e^{-j\frac{\pi m(2N-1)}{2N}}e^{j\frac{\pi mn}{2N}}\right]. \tag{27}$$

For $m = 2k + 1$, simplify the exponential term $e^{-j\frac{\pi(2k+1)(2N-1)}{2N}} = e^{-j\pi(2k+1)}e^{j\frac{\pi(2k+1)}{2N}} = -e^{j\frac{\pi(2k+1)}{2N}}$, considering $e^{-j\pi(2k+1)} = -1$). Substitute into Eq.(27):

$$X_{\text{sym}}[2k + 1] = -\sum_{n=0}^{N-1} x[n]\left[e^{j\frac{\pi(2k+1)n}{2N}} + e^{-j\frac{\pi(2k+1)n}{2N}}\right]. \tag{28}$$

Comparing Eq.(25) and Eq.(28), the summation in Eq.(28) is exactly $2X_C[k]$. Rearranging gives the key equivalence:

$$X_C[k] = -\frac{1}{2} \cdot \text{DFT}\{x_{\text{sym}}[n]\}\bigg|_{m=2k+1}. \tag{29}$$

This formula shows the DCT-II is uniquely determined by the DFT of the even-symmetric extended signal. No approximations or ad-hoc rules are involved in the derivation. Thus the theorem holds. $\square$

