# OpenReview forum: "Winformer: Transcending Pairwise Similarity for Time-series Generation"
_ICML.cc/2026/Conference — ICML 2026 regular_

### Official Review · Reviewer_TgXN · 2026-03-06

**Soundness:** 3
**Presentation:** 3
**Significance:** 3
**Originality:** 3
**Overall Recommendation:** 4
**Confidence:** 3

**Summary:**

This paper proposes Winformer, a novel diffusion framework built on a window-wise attention mechanism that transcends pairwise point similarity. Winformer’s key components include: (1) Ample Attention, which replaces pairwise similarity with window-wise comparison via learnable convolutions initialized with DCT bases for efficiency; (2) a hybrid encoder block fusing Ample Attention and vanilla attention to balance window-wise and point-wise features; (3) domain conditioning with prompts to facilitate cross-domain generation.Extensive experiments on 12 real-world datasets (spanning energy, weather, transportation, economics) demonstrate that Winformer outperforms 6 baselines (GAN/VAE/diffusion-based) by an average of 10.67% in MMD.

**Compliance With Llm Reviewing Policy:**

Affirmed.

**Key Questions For Authors:**

Please see weaknesses

**Limitations:**

yes

**Strengths And Weaknesses:**

Strengths:
1.The work offers originality through multiple innovations (e.g., Ample Attention, hybrid encoder).
2.The work is technically rigorous with solid theoretical and empirical support.
3.The paper is clearly written and well-structured, following a logical flow from problem identification to theoretical derivation, model design, and experimental validation.

Weaknesses:
1.The application scenarios of time series generation seem less intuitive than those of tasks such as time series forecasting and time series classification. Are there any specific applications of time series generation? Could the authors elaborate on the application scenarios of time series generation in detail?
2.Winformer performs less effectively on trend-dominant datasets (e.g., Wind, Temperature)—can the framework be augmented with a lightweight trend modeling component (e.g., Fourier feature encoding for trends) without compromising periodicity alignment?
3.Winformer relies on explicit periodicity—how does it perform on non-periodic time-series datasets?
4.What is the number of experimental runs to generate the results in Table 1?
5.It is recommended that the captions for Figures 4-12 be more detailed to facilitate readers' understanding.

---

> ### Author Rebuttal · Authors · 2026-03-28
>
> > **Complete tables for more datasets** are in: https://anonymous.4open.science/r/rebuttal_tables_for_Winformer/Tables.pdf
>
> ---
>
> **[W1] specific applications of time series generation**
>
> Additional results show our practical value in Table 18 and 19. These applications shows how the generated data enhance the downstream models when the existing data is limited, such as classification and forecasting, by training on  on synthetic data and testing on real data (TSTR). We provide the results for classification and forecasting as you recommended. The predictor and classifier are vanilla transformer, and our method could help them to achieve better performance.
>
>   Table 18: Results on Application (forecasting).
>
>   |       | TSTR(Ours) |        | Original |        |
>   | ----- | ---------- | ------ | -------- | ------ |
>   |       | MSE(↓)     | MAE(↓) | MSE(↓)   | MAE(↓) |
>   | ETTh1 | 0.7411     | 0.6727 | 0.7682   | 0.6898 |
>   | ETTh2 | 1.3013     | 0.9295 | 1.3159   | 0.9352 |
>   | ETTm1 | 0.4974     | 0.5013 | 0.4997   | 0.502  |
>   | ETTm2 | 0.3881     | 0.4851 | 0.3785   | 0.4742 |
>   | ECL   | 0.2435     | 0.3432 | 0.2552   | 0.3481 |
>
>   Table 19: Results on application for classfication (Accuracy, ↑).
>
>   |                     | TSTR(Ours)   | Original |
>   | ------------------- | ------ | -------- |
>   | FaceDetection       | 0.6827 | 0.6765   |
>   | Heartbeat           | 0.7756 | 0.7707   |
>   | JapaneseVowels      | 0.9757 | 0.9784   |
>   | PEMS-SF             | 0.8382 | 0.8324   |
>   | SpokenArabicDigits  | 0.9859 | 0.9845   |
>   | UWaveGestureLibrary | 0.8625 | 0.8594   |
>
> ---
> **[W2]  Augmenting with a lightweight trend modeling component**
>
> To improve on trend-dominant datasets, our model is able to integrate trend analysis module (Autoformer style), with results shown in Table 20,  showing our model's powerful performance with basic existing trend analysis. Our design does not prevent trend extraction, and it's not central to the main focus of our paper.
>
>   Table 20: Integrating Winformer with Trend decomposition (MMD, ↓)
>
>   |             | Winformer | Winformer + Trend |
>   | ----------- | --------- | ----------------- |
>   | Wind        | 0.034     | 0.030             |
>   | Fred-MD     | 0.002     | 0.002             |
>   | Temperature | 0.230     | 0.202             |
>
> ---
>
> **[W3] Non-periodicity time-series data**
>
> We can improve the approach by incorporating a simple trend analysis as W2. We also want to clarify that real-world time series frequently exhibit periodic patterns because they are inherently driven by recurring natural phenomena and repetitive human social and economic activities, a well-recognized characteristic in time series analysis[1]. Perfectly non-periodic sequences are uncommon in real world data.
>
> ---
> **[W4] Experimental runs.**
>
> Experiments were repeated 5 times with seeds spanning 2021–2025.
>
> ---
> **[W5] More detailed captions**
>
> Thank you for your advice! We will provide more detailed captions for Figure 4-12 in future versions. (As excessive text in linked images is temporarily not allowed during rebuttal which probably to be judged as cheating (ICML rules), we can only make detailed revisions after the review is approved.)
>
> ---
>
> [1] George E. P. Box, Gwilym M. Jenkins. Time Series Analysis: Forecasting and Control, 1970.

---

> > ### Author Rebuttal · Reviewer_TgXN · 2026-04-06
> >
> > Thanks for your response. As my original score is positive, I'll keep my score.

---

### Official Review · Reviewer_CeTZ · 2026-03-12

**Soundness:** 3
**Presentation:** 3
**Significance:** 3
**Originality:** 3
**Overall Recommendation:** 4
**Confidence:** 3

**Summary:**

The paper introduces Winformer, a novel diffusion framework designed for cross-domain time-series generation. The authors argue that existing point-wise and patch-wise attention mechanisms struggle with periodicity misalignment across different domains because they either lose or artificially break continuous periodic patterns. To resolve this, Winformer introduces a "window-wise attention mechanism" that replaces standard pairwise point similarity with continuous window comparisons. By applying frequency decomposition (via a reduced Discrete Cosine Transform kernel), the model learns adaptive window-alignment kernels that effectively capture and transfer complex periodic patterns. Winformer is evaluated on 12 real-world datasets across four domains, demonstrating an average 10.67% improvement in Maximum Mean Discrepancy (MMD) compared to state-of-the-art diffusion baselines like TimeDP.

**Compliance With Llm Reviewing Policy:**

Affirmed.

**Final Justification:**

The authors appear to discuss the key issue of periodicity misalignment more concretely, and overall, the authors explore a relevant challenge with stronger empirical support after the rebuttal.

The rebuttal improves the paper; e.g., adding medical datasets, expanded statistical testing, and controlled experiments isolating the alignment mechanism (with a fixed diffusion backbone) strengthen the soundness of the claims. The clarification on multivariate settings without GCN and the revised positioning of DCT are also helpful.

While some concerns remain (the magnitude of gains, limited diversity of attention baselines), the overall empirical validation is now more convincing.
Given these improvements, I update my assessment.

**Key Questions For Authors:**

1. Can you provide statistical significance tests (e.g., paired t-tests across different seeds) for the improvements over TimeDP? Many results in Table 1 have overlapping confidence intervals.

2. The stride length s=25 is chosen to exceed common periodic cycles, but what happens when the true periodicity is much larger than the kernel size (e.g., monthly or seasonal patterns in longer sequences)? Is there a principled way to set this hyperparameter, perhaps adaptively per domain?

3. How does Winformer perform on multivariate time-series generation (i.e., generating all channels jointly rather than univariate slices)? The current evaluation on univariate slices may not fully capture cross-channel dependencies.

4. Could you include downstream utility evaluations, such as training a classifier or forecaster on synthetic data and testing on real data (TSTR), to demonstrate practical value beyond distributional metrics?

5. The visualization in Figure 6 suggests that randomly initialized kernels converge to approximately uniform weights (similar to average pooling). Does this imply that the DCT initialization is not strictly necessary, or does it primarily help with convergence speed?

**Limitations:**

Authors do not provide a dedicated Limitations section in the main text.

**Strengths And Weaknesses:**

Strengths:

1. The core idea of shifting from pairwise point attention to a window-to-window comparison using frequency decomposition is good.

2. Domain confusion caused by phase and periodicity misalignment is a very real problem in cross-domain time-series generation.

3. The theoretical foundation is well-supported.

Weaknesses:
1. While the evaluation covers 12 datasets across energy, weather, transportation, and economics, it completely omits healthcare time series. Medical data (such as ECGs or continuous physiological monitors) often exhibit highly complex, multi-scale periodicities that are frequently interrupted by anomalies or irregular sampling. It is unclear if the proposed rigid window-wise alignment holds up in these significantly noisier domains.

2. The paper emphasizes "transcending patching" and "overlapping windows," but the chosen baselines are primarily generation-focused models (TimeGAN, TimeVAE, TimeDP, Diffusion-TS). It lacks empirical comparisons against other specialized time-series attention variants that also modify patching or use windowed attention (e.g., FWin, Swin-based architectures, or dynamic patching mechanisms), making it difficult to isolate the benefit of the specific Ample attention design from the diffusion framework itself.

---

> ### Author Rebuttal · Authors · 2026-03-28
>
> > **Complete results for following tables** are in: https://anonymous.4open.science/r/rebuttal_tables_for_Winformer/Tables.pdf
> ---
> **[W1] Adaptation on Medical Data**
>
> Our model outperforms on PTBXL (Table 11), an ECG dataset as you recommended. DCT provides a frequency-selective inductive bias, because DCT concentrates structured periodic energy into a small set of coefficients, emphasizing low-to-mid frequency components can improve robustness to certain artifacts.
>
> Table 11:  Results on ECG data
> ||KL(↓)|MMD(↓)|
> |-|-|-|
> |Ours|1.424|0.3496|
> |TimeDP|1.552|0.3532|
> |DiT|1.446|0.3512|
>
> ---
> **[W2] Comparison against attention variants**
>
> Our model wins in more datasets against other attention variants (FWin and Patching variants) in Table 12. Unlike FWin (window attention followed by Fourier mixing), our method injects frequency decomposition into the alignment stage, which is designed for cross-domain periodicity misalignment, achieving better performance compared to other baselines.
>
> Table 12: Results for attention variants (MMD, ↓)
>
> ||window-align (ours)|patch-align|point-align|FWin|Swin|
> |-|-|-|-|-|-|
> |Electricity|0.001|0.002|0.002|0.001|0.001|
> |Air|0.011|0.034|0.012|0.014|0.045|
> |Rain|0.036|0.052|0.039|0.052|0.045|
> |Exchange|0.137|0.139|0.139|0.139|0.148|
>
> ---
> **[Q1] Significance tests**
>
> Following your suggestion,  we test the significance on MMD across 5 seeds with paired T test (in Table 13). With low p-value (<0.05) (Ours vs. TimeDP), following results confirm our model's consistent lead over TimeDP is statistically reliable across diverse datasets.
>
> Table 13: Paired T-Test across seeds.
> |Dataset|t_value|p_value|
> |-|-|-|
> |Electricity|-7.052|0.002|
> |Air|-6.645|0.003|
> |Exchange|-6.116|0.004|
> |Rain|-6.025|0.004|
>
> ---
> **[Q2] Adaptive to larger cycles or different domains**
> - (1) Adaptive to larger cycles: The hyper-parameter s can be set to a large enhough value. And we can provide some basical cycle estimation method for time-series data, such as ACF. Notably, excessively large periods may be meaningless. As recommend in [1], a minimum of 3–5 cycles is required to identify a periodic pattern with any reliability. Thus, we recommend $s\leq \frac{L}{5}$ for a better performance.
>
> - (2) Adaptive to different domains: This parameter could **backward compatible with smaller cycles**. E.g. s=25 contains cycle sizes 1,2,...25 as Eq(11) states. So, set the s as the largest estimated cycle size of different domains will make it better. If there's difficulty in estimating the periodicity, set $s=\frac{L}{5}$ usually can cover most of the common cycles.
> ---
> **[Q3] Multivariate time-series generation**
>
> Our model outperforms baseline on PEMS datasets (multivariate time-series datasets with complex cross-channel dependencies) in Table 14. We just inject the cross-domain dependencies (captured by a GCN) as condition to our model, and our model can realize the channel jointly generation with the dependencies.
>
> Table 14: Results on PEMS datasets.
> ||TimeDP||Ours||
> |-|-|-|-|-|
> ||KL(↓)|MMD(↓)|KL(↓)|MMD(↓)|
> |PMES03|1.67|0.34|1.53|0.29|
> |PMES04|2.29|0.33|1.88|0.21|
> |PMES08|1.80|0.34|1.61|0.28|
>
> ---
> **[Q4] Downstream Evaluation**
>
> Additional results show our practical value in Table 15 and 16, for classification and forecasting as you recommended. The predictor and classifier are vanilla transformer, and our method can help them to achieve better or comparable performance.
>
> Table 15: Application for forecasting.
> ||TSTR(Ours)||Original||
> |-|-|-|-|-|
> ||MSE(↓)|MAE(↓)|MSE(↓)|MAE(↓)|
> |ETTh1|0.7411|0.6727|0.7682|0.6898|
> |ETTh2|1.3013|0.9295|1.3159|0.9352|
> |ETTm1|0.4974|0.5013|0.4997| 0.502|
> |ETTm2|0.3881|0.4851|0.3785|0.4742|
> |ECL|0.2435|0.3432|0.2552|0.3481|
>
> Table 16: Application for classfication (Accuracy, ↑).
> ||TSTR(Ours)|Original|
> |-|-|-|
> |FaceDetection|0.6827|0.6765|
> |Heartbeat|0.7756|0.7707|
> |JapaneseVowels|0.9757|0.9784|
> |PEMS-SF|0.8382|0.8324|
> |SpokenArabicDigits|0.9859|0.9845|
> |UWaveGestureLibrary|0.8625|0.8594|
>
> ---
> **[Q5] Advantage for DCT**
>
> Q5: DCT initialization is functionally necessary, not just for speed. First, Table 17 shows that random initialization yields inferior performance even after convergence, proving it fails to fully capture the precise spectral priors DCT provides. Second, DCT offers significant efficiency, saving ~5,000 training steps (\~1 hour) by starting from an informed frequency-aware state.
>
> Table 17: Initialization Comparison (MMD, ↓)
> ||Avg Pool|random conv|DCT Conv|
> |-|-|-|-|
> |Electricity|0.002|0.001|0.001|
> |Air|0.012|0.014|0.011|
> |Rain|0.036|0.048|0.036|
> |Exchange|0.138|0.137|0.137|
>
> ---
>
> Limitation: Thank you. We will add a Limitations section, discussing hyperparameter sensitivity for long seasonal cycles, and multivariate dependency evaluation beyond marginal metrics.
>
> ---
> [1] George E. P. Box, Gwilym M. Jenkins. Time Series Analysis: Forecasting and Control, 1970.

---

> > ### Author Rebuttal · Reviewer_CeTZ · 2026-04-01
> >
> > The authors appear to discuss the key issue of periodicity misalignment more concretely in the rebuttal, and Overall, the authors explore a relevant challenge with a clearer empirical backing than in the original submission.
> >
> > That said, several core concerns are only partially addressed:
> >
> > 1. The added ECG result is useful but weak. The margin over baselines is small (Table 11), and one dataset is not enough to support claims about robustness to noisy, irregular medical data. The concern about irregular sampling and anomalies remains largely untested.
> > 2. Comparison to attention variants: The added comparison (Table 12) is appreciated, but still limited. The evaluation is narrow (few datasets, only MMD), and does not convincingly isolate whether gains come from the alignment mechanism vs. diffusion backbone.
> > 3. The t-test helps, but is reported on a small subset of datasets. Given overlapping confidence intervals in Table 1, a more comprehensive analysis would be expected.
> > 4. Multivariate setting: The PEMS results are a positive addition, but rely on external GCN conditioning. It is still unclear whether Winformer itself captures cross-channel dependencies or depends on auxiliary structure.
> > 5. DCT necessity: The new ablation suggests benefits, but improvements are marginal in some cases. The claim of “functional necessity” feels overstated.
> >
> > Overall, the rebuttal improves the paper and addresses several questions, but key concerns about generality, isolation of contributions, and robustness remain only partially resolved.

---

> > > ### Author Response · Authors · 2026-04-03
> > >
> > > # Response to Reviewer CeTZ
> > > We sincerely thank the reviewer for recognizton and feedback. We will further address each concern in detail below.
> > >
> > > ---
> > >
> > > **[Q1] Robustness on irregular / noisy data**
> > >
> > > First, our original 12 datasets already include several with inherently noisy and non-stationary dynamics. While these are regularly sampled, their temporal patterns are highly irregular in nature, partially reflecting real-world noise conditions.
> > >
> > > - **Financial datasets** (such as Exchange) often exhibit irregular patterns due to missing or delayed recordings, leading to uneven temporal dynamics and noisy observations.  Our model improves ~10\% over SOTA.
> > > - **Weather datasets**(such as rain) contain inherent noise and irregular fluctuations, especially under extreme weather. Our model improves ~30\% over SOTA.
> > >
> > > Second, to directly address the concern on irregular medical data, we now evaluate on **three additional medical datasets**. As shown in Table 21, Winformer consistently outperforms both TimeDP and DiT across all three datasets on both metrics.
> > >
> > > Table 21 results for medical datasets
> > >
> > > ||Ours||TimeDP||DiT||
> > > |-|-|-|-|-|-|-|
> > > |Metric|KL|MMD|KL|MMD|KL|MMD|
> > > |PTB-XL|1.424|0.350|1.552|0.353|1.446|0.351|
> > > |illness|1.628|0.598|1.645|0.671|1.637|0.626|
> > > |Health|1.502|0.417|1.685|0.438|1.613|0.428|
> > >
> > > Overall, our experiments now span **15 datasets** (12 original submission, 3 added during rebuttal) with diverse characteristics, providing stronger empirical support for robustness, which directly addresses the concern.
> > >
> > > ---
> > >
> > > **[Q2] Contribution isolation (alignment vs. diffusion backbone)**
> > >
> > > To isolate the effect of the alignment mechanism, **all comparisons are conducted under a fixed diffusion backbone (DiT)**. We replace only the attention mechanism for each attention variant while keeping all other components (architecture, training schedule, hyperparameters) unchanged.
> > >
> > > This controlled design ensures that the performance gains are not attributable to the diffusion backbone, but to the alignment mechanism itself. As results shown in **supplementary link**: https://anonymous.4open.science/r/rebuttal_tables_for_Winformer/table_22.pdf, our method achieves the best performance in **18 out of 24** entries and consistently outperforms all alternative attention variants on both KL and MMD. Since the backbone is held constant, the observed gains are attributable to the alignment mechanism.
> > >
> > > ---
> > >
> > > **[Q3] Statistical significance**
> > >
> > > We provide **full statistical t-test results across 12 datasets** in the **supplementary link**: https://anonymous.4open.science/r/rebuttal_tables_for_Winformer/table_23.pdf
> > >
> > > Our method achieves statistically significant improvements in most cases, supporting that the gains are consistent rather than incidental.
> > >
> > > Regarding the overlapping confidence intervals in Table 1: overlapping CIs do not necessarily imply non-significance. It is well established that two 95\% confidence intervals can overlap by up to approximately 50\% while the corresponding t-test still rejects the null hypothesis at p < 0.05. Our t-test results confirm that the observed performance gaps are statistically meaningful despite visual CI overlap.
> > >
> > > ---
> > >
> > > **[Q4] Multivariate modeling (generality)**
> > >
> > > In the original PEMS experiments, GCN is used as a structural prior following common practice, and was added to all baselines for fair comparison. To directly address the reviewer’s concern, we additionally evaluate **without GCN conditioning**.
> > >
> > > Winformer still consistently outperforms strong baselines in this setting, demonstrating it **does not rely on external structures** to perform better on cross-channel cases than other baseline, with results in Table 24.
> > >
> > > Table 24 results on PEMS dataset
> > >
> > > ||TimeDP||Ours||
> > > |-|-|-|-|-|
> > > |Metric|KL|MMD|KL|MMD|
> > > |PEMS03|1.73|0.38|1.66|0.31|
> > > |PEMS04|2.50|0.37|2.00|0.32|
> > > |PEMS08|1.99|0.39|1.71|0.32|
> > >
> > > The performance remains comparable to those with GCN-based setting. This demonstrates that Winformer's gains are **intrinsic to the model** and do not rely on external structural priors. GCN is an optional enhancement rather than a prerequisite.
> > >
> > > ---
> > >
> > > **[Q5] The role of DCT**
> > >
> > > We agree that the phrase "functional necessity" overstates the claim and will revise the wording in the manuscript.
> > >
> > > To clarify our intended position: DCT initialization is not strictly required for the framework to function—as shown in the ablation, the AvgPool/RConv variant already performs competitively. However, DCT initialization consistently provides additional gains by enabling the model to leverage higher-frequency basis  from the start, which improves training convergence and final performance, particularly on datasets with rich periodicity. We will reframe this as a **recommended default** that offers consistent practical benefits, rather than a strict necessity.
> > >
> > > ---
> > > We sincerely appreciate the reviewer's engagement. We hope these additional analyses and clarifications fully address the reviewer's remaining concerns.

---

### Official Review · Reviewer_oQ6K · 2026-03-13

**Soundness:** 3
**Presentation:** 3
**Significance:** 2
**Originality:** 3
**Overall Recommendation:** 4
**Confidence:** 4

**Summary:**

This paper proposes Winformer, a diffusion-based framework for cross-domain time-series
generation. The core innovation lies in shifting the fundamental processing unit of the
attention mechanism from point-to-point similarity computation to window-wise comparison.
Through theoretical derivation grounded in the Discrete Fourier Transform (DFT) and the
convolution theorem, the authors reduce window-wise alignment to a learnable 2D convolutional
kernel initialized with DCT bases, termed "Ample Attention." This module is concatenated
in parallel with standard vanilla attention to form a hybrid encoder. Experiments on 12
real-world datasets demonstrate an average MMD improvement of approximately 10.67% over
baselines including TimeDP and Diffusion-TS.

**Compliance With Llm Reviewing Policy:**

Affirmed.

**Final Justification:**

The authors solved my concerns, and I changed the rating from 3 to 4

**Key Questions For Authors:**

## Key Questions For Authors

**Q1: How do you explain the K-L regression?**
On Wind (0.202 vs. 0.152), Traffic, and Fred-MD, Winformer's K-L divergence is notably
worse than TimeDP, contradicting the MMD trend. What does this inconsistency indicate? Does
it suggest that Winformer-generated sequences exhibit distributional bias in local patterns?

**Q2: How generalizable is the stride length choice?**
The stride s=25 is chosen to "cover the common period of 24," but datasets have vastly
different sampling rates (e.g., Wind at 4-second intervals vs. Fred-MD at monthly intervals).
Is a single fixed stride truly appropriate across all domains? Have adaptive stride strategies
been explored?

**Q3: What is the essential distinction from LocalMHA / Convolutional Attention?**
Could the authors include an ablation directly comparing (a) randomly initialized Conv2d,
(b) a standard local window attention (e.g., Longformer-style), and (c) Ample Attention
with DCT initialization, to more precisely isolate the contribution of the frequency-domain
derivation and DCT initialization?

**Q4: What is the isolated contribution of domain conditioning?**
If domain prompts are removed (leaving only step embeddings), how does performance change?
What fraction of the reported 10.67% MMD improvement is attributable to window alignment
versus domain conditioning?

**Q5: What is the plan for trend-dominated data?**
For datasets such as Wind and Temperature where trend components dominate, do the authors
plan to incorporate trend-cycle decomposition preprocessing, or to adaptively downweight
window alignment, in order to improve the method's generality?

**Limitations:**

## Limitations

The authors briefly mention future work directions in the Conclusion, but the paper
**lacks a dedicated Limitations section**. The following important limitations are not
acknowledged:

1. **Undefined applicability boundary**: The method degrades on trend-dominated data
   (Wind, Temperature), yet no criterion is provided to guide practitioners on when
   Winformer should or should not be used.

2. **Fixed sequence length**: All experiments use sequences of length 168. The method's
   generalization to different sequence lengths is entirely unverified.

3. **Computational resource requirements**: Experiments are conducted on a V100 32GB GPU;
   applicability to resource-constrained settings is not discussed.

4. **Univariate-only setting**: Datasets are pre-processed into univariate sequence slices.
   The extension of window-wise alignment to multivariate time-series — a common real-world
   requirement — is not explored.

**Strengths And Weaknesses:**

## Strengths

1. **Well-motivated with clear intuition**
   The two core failure modes — point-wise modeling losing periodic structure and fixed
   patching breaking evolving cycles — are clearly illustrated with figures (Figure 1, 13),
   making the problem formulation easy to follow.

2. **Theoretically coherent derivation**
   The derivation chain from DFT window similarity, through the convolution theorem to a
   learnable convolutional kernel, and further justified via Parseval's theorem (Section 4.1,
   Appendix G), is complete and self-consistent. The formal proof of DCT as a real-valued
   reduction of DFT (Lemma G.1) is particularly rigorous.

3. **Solid experimental protocol**
   The authors follow the same evaluation protocol as TimeDP, repeat all experiments five
   times with different random seeds and report standard deviations, and conduct ablation
   studies comparing point-wise, patch-wise, and window-wise alignment — lending credibility
   to the results.

4. **Convincing visualization analysis**
   The attention heatmap analysis on synthetic data (Figure 5) and the kernel visualization
   before and after training (Figure 6) provide intuitive, interpretable support for the
   design choices and enhance the paper's explainability.

## Weaknesses

**W1: Marginal overall improvement overstated**
The advertised "10.67% MMD improvement" is driven by large gains on a small subset of
datasets. On Electricity, Air, and Fred-MD, Winformer and TimeDP produce nearly identical
MMD values (e.g., both 0.001, 0.011, 0.002) within each other's standard deviation. More
critically, Winformer wins fewer datasets than TimeDP on the K-L divergence metric
(approximately 7 vs. 17 out of 24), indicating that improvements across the two metrics
are inconsistent. This discrepancy deserves deeper analysis rather than selective emphasis
on the MMD average.

**W2: Methodological contribution is incremental**
The final form of Ample Attention reduces to applying a learnable 2D convolution over the
standard attention score matrix, which bears close resemblance to existing locally enhanced
Transformer designs (e.g., LocalMHA, ConvAttention). Furthermore, Table 3 shows that the
simple Avgpool kernel (plain mean convolution) achieves 18 best counts versus 25 for the
DCT-initialized Conv2d kernel, suggesting that the specialized DCT initialization is not
the decisive factor and the unique contribution of the method needs further clarification.

**W3: Degradation on trend-dominated data lacks analysis**
Winformer underperforms the baseline on Wind and Temperature — datasets with stronger
trend components — and the authors dismiss this with a single sentence citing "trendiness."
No systematic analysis is provided on why window-wise alignment is harmful for non-periodic
sequences, and no mitigation strategies are proposed, leaving the method's applicability
boundary undefined.

**W4: Computational cost evaluation is incomplete**
Table 4 reports only batch throughput without memory usage or parameter count comparisons.
The theoretical complexity increases from O(Lx²D) to O(s²·Lx²D), which at s=25 represents
a substantial multiplicative factor. Yet the observed throughput reduction is only ~21%
(2.92→2.30 batch/s). The gap between theoretical complexity and empirical overhead is never
explained, which undermines the authors' claim that the additional cost is "acceptable."

**W5: Contribution of domain conditioning is entangled**
Section 4.3's domain conditioning design is largely inherited from TimeDP's domain prompt
framework with minimal modification. The paper lacks a two-factor ablation (with/without
window alignment × with/without domain conditioning), making it impossible to attribute the
performance gains to specific components.

**W6: Baselines are not sufficiently up-to-date**
TimeGAN (2019), GT-GAN (2022), and TimeVAE (2021) are relatively dated. Several more recent
diffusion-based models mentioned in the Related Work — including TimeDiT, UTSD, and T2S
(2024–2025) — are not included in the comparison, weakening the validity of claims about
achieving state-of-the-art performance.

---

> ### Author Rebuttal · Authors · 2026-03-28
>
> > **Complete results for following tables**: https://anonymous.4open.science/r/rebuttal_tables_for_Winformer/Tables.pdf
>
> ---
> **[W1] Metric discrepancy**
> We‘d like to clarify two factual points, followed by metric analysis.
> - Clarification A: The 10.67% MMD improvement is the average across all datasets, not a selected subset. The MMD values in Electricity and Fred-MD is lowest (0.001,0.002) among all datasets, they are near-optimal. We preserve them in average improvement for fair comparison.
> - Clarification B: Winformer wins **more** than TimeDP, outperforming on **19 out of 24** lines (Page 6, Table 1), wins 10/12 in MMD and 9/12 in KL.
> - Metric discrepancy from its design. MMD measures global distribution, while KL is sensitive to local point-wise noise. Our Ample attention improves periodic modeling, leading to stronger gains on global metrics such as MMD.
>
> ---
> **[W2] Methodological contribution**
> - Clarification on contribution: We establish a **frequency-domain** derivation for the integration of convolution over attention and the final form is a special implementation, while LocalMHA & ConvAttention are heuristic designs explained by our derivation. We provide results as you recommended in Q3 (Table 7), showing our advancement in a full frequency modeling.
>
> Table 7: Ablation for windows (↓)
> ||random_conv|Longformer|Ours|
> |-|-|-|-|
> |Air|0.014|0.015|0.011|
> |Rain|0.048|0.040|0.036|
> |Exc|0.137|0.142|0.137|
> - We emphasize that AvgPool is not an external baseline but a special case within our proposed framework. As Eq(11), DCT contains $\psi(0), \psi(1),... \psi(p-1)$ and AvgPool contains $\psi(0)$ only. Therefore, the fact that AvgPool achieves 18 best counts **validates the effectiveness of our framework**. And DCT shows progressive improvement with full spectrum, demonstrating the DCT initialization is a contributing factor.
> ---
> **[W3] Analysis for Trend-dominated data**
> - We'd like to clarify that we uses Z-Score normalization to remove stationary trends. Residual trends in Wind/Temp are slow, large-scale drifts longer than our window size. This is a known scope constraint of periodicity-oriented designs, not harmful to trends.
> - Mitigation strategy. We integrate Winformer with a basic trend adaptation (Autoformer-style)  in Table 8, which will be added to the final version.
>
> Table 8: Integrating with Trend adaptation (↓)
> ||Ours|Ours+Trend|
> |-|-|-|
> |Wind|0.034|0.030|
> |Fred-MD|0.002|0.002|
> |Temp | 0.230|0.202|
>
> ---
> **[W4] Computational Cost**
> - Model size: A 6-layer DiT has \~29.3M parameters; Winformer only adds \~2k parameters for the kernel.
> - Complexity vs runtime. The theoretical complexity O(s²·Lx²D) applies **only to Ample Attention Layer**, while the empirical results (2.30 batch/s) reflects the **entire pipeline** (preprocess, optimization,...). Since the overall pipeline dominates runtime, the extra overhead of Ample Attention is acceptable.
> ---
> **[W5] Domain Conditioning**
> - Differences from TimeDP: TimeDP injects prompts via attention masking, while we use adaptive layer with multi-cond instead, avoiding interference with periodicity extraction in attention.
> - As suggested, we provide results in Table 9, verifying effectiveness of both components. Partial related ablation is also included in Table 2 (Page 7), where WO window alignment = point-wise alignment.
>
> Table 9: Ablation study (↓)
> ||Ours|WO window align|WO domain condition|WO both|
> |-|-|-|-|-|
> |Air|0.011|0.012|0.030|0.031|
> |Rain|0.036|0.039|0.192|0.189
> |Exc|0.137|0.139|0.140|0.138|
>
> ---
> **[W6] Baselines**
>
> - Clarification on baseline: Our paper is not limited to dated methods. TimeDP (AAAI 2025) and Diffusion-TS (ICLR 2024) are recognized 2024-2025 SOTA.
> - The recommended baselines: T2S targets text-conditioned series generation, we believe to be out-of-scope. For TimeDiT (KDD workshop 2025) and UTSD (arxiv 2024), neither has released an official implementation, nor do they report performance gains over TimeDP (2025 SOTA).
> - Nevertheless, our best-effort reproduction confirms that Winformer maintains its lead (Table 10).
>
> Table 10: Comparison to new baselines (↓)
> ||Ours|TimeDiT|UTSD|
> |-|-|-|-|
> |Air|0.011|0.018 |0.017|
> |Rain|0.036|0.040|0.037|
> |Exc|0.137|0.170|0.169|
>
> ---
> **[Q1] KL metric**
>
> We address the metric-level analysis in W1 and provide dataset-specific clarifications here.
> - Clarification: Winformer actually outperforms TimeDP on Fred-MD (0.201 vs. 0.203), and equal on Traffic (0.002 vs. 0.002).
> - As discussed in W3, KL on Wind penalizes more heavily than MMD. KL reflect the metric's sensitivity to small noise, our proposed Winformer could alleviate noise spreading by windowed operation.
>
> ---
> **[Q2] Stride length**
>
> Stride s=25 covers common periodicities (≤25) and adaptively captures diverse periodicities (Sec 5.1 (d)), e.g. Fred-MD contains monthly records with a cycle of 12 (a year) < 25. Thus, though the stride is fixed, it could adap to diverse dataset.
>
> ---
> - Q3: In W2
> - Q4: In W5
> - Q5: In W3

---

### Decision · Program_Chairs · 2026-04-30

**Decision:**

Accept (regular)

**Comment:**

The paper addresses an important problem in cross-domain time-series generation, and reviewers generally agreed that the motivation, technical development, and presentation are strong. The proposed window-wise alignment mechanism was viewed as a meaningful idea, supported by a solid derivation and overall positive empirical results.

The main concerns were about the breadth of evaluation, the consistency of gains across settings and metrics, and the isolation of contributions from other components. After the rebuttal, these concerns were substantially addressed through added experiments, stronger controlled comparisons, significance testing, and clarifications on medical, multivariate, and downstream settings. Some limitations remain, but they do not outweigh the paper's strengths.

I have read the paper, rebuttal, and discussion carefully. Overall, I find the paper technically sound, novel enough, and useful to the community. I recommend weak acceptance.